# Ventral pallidum GABA and glutamate neurons drive approach and avoidance through distinct modulation of VTA cell types

Lauren Faget [1,2] ✉, Lucie Oriol[1], Wen-Chun Lee [1], Vivien Zell [1], Cody Sargent[1], Andrew Flores[1,2], Nick G. Hollon [3], Dhakshin Ramanathan[2,3] & Thomas S. Hnasko [1,2] ✉

The ventral pallidum (VP) contains GABA and glutamate neurons projecting to ventral tegmental area (VTA) whose stimulation drives approach and avoidance, respectively. Yet little is known about the mechanisms by which VP cell types shape VTA activity and drive behavior. Here, we found that both VP GABA and glutamate neurons were activated during approach to reward or by delivery of an aversive stimulus. Stimulation of VP GABA neurons inhibited VTA GABA, but activated dopamine and glutamate neurons. Remarkably, stimulation-evoked activation was behavior-contingent such that VTA recruitment was inhibited when evoked by the subject's own action. Conversely, VP glutamate neurons activated VTA GABA, as well as dopamine and glutamate neurons, despite driving aversion. However, VP glutamate neurons evoked dopamine in aversion-associated ventromedial nucleus accumbens (NAc), but reduced dopamine release in reward-associated dorsomedial NAc. These findings show how heterogeneous VP projections to VTA can be engaged to shape approach and avoidance behaviors.

The ventral pallidum (VP) is a primary output of the nucleus accumbens (NAc), provides direct input to both ventral tegmental area (VTA) and lateral habenula (LHb)[1–3], and thus lies at a privileged anatomical position to modulate motivated behaviors[4–9]. Rats will work to electrically self-stimulate VP[10], pharmacological activation of the VP can trigger feeding in sated animals[11,12], while VP lesions induce aphagia and cause rats to show aversive reactions to sucrose[13]. Moreover, in vivo recordings of neural activity suggest that VP cells encode hedonic stimuli, reward prediction, and goal-directed behaviors[14–21]. VP also plays critical roles in drug self-administration, extinction, and reinstatement of drug-seeking behaviors[22–29]. While these studies reveal that VP can profoundly shape reward processes, we have only recently begun to appreciate the distinct functional roles of diverse VP cell types.

VP is highly heterogeneous[3,7,30,31]. Though canonically considered GABAergic, some VP neurons express cholinergic or glutamate (Glut) markers[7,32,33]. Glut and GABA neurons in VP share highly similar patterns of projection, with particularly dense projections to VTA and LHb, whereas cholinergic neurons display distinct projection patterns[33–36]. Optogenetic manipulation of VP GABA and Glut cell bodies, or their VTA-projecting terminals, lead to opposite effects on reinforcement assays; VP GABA neurons drive reward or approach responses, and VP Glut neurons drive avoidance[33–35,37]. Both VP GABA and Glut neurons make functional synapses onto diverse cell types in

[1]Department of Neurosciences, University of California San Diego, La Jolla, CA, USA. [2]Research Service, Veterans Affairs San Diego Healthcare System, San Diego, CA, USA. [3]Department of Psychiatry, University of California, San Diego, La Jolla, CA, USA. ✉e-mail: lfaget@health.ucsd.edu; thnasko@health.ucsd.edu

VTA, but only activation of VP GABA neurons induces a significant Fos expression in VTA DA neurons, pointing to polysynaptic disinhibition of DA cells[33,34]. These findings strongly support cell-type and projection-specific roles for VP neurons in processes underlying behavioral reinforcement and, importantly, drug addiction[38–42].

To understand how VP cell types drive opponent patterns of behavior, we sought to determine how activity in VP GABA and Glut neurons changes in response to rewarding and aversive stimuli, and how these VP cell types differentially control the activity of diverse VTA cell types. We first used calcium sensors and fiber photometry to record activity of VP cell types in response to appetitive and aversive stimuli. We then investigated how VP cell type activity modulated activity in VTA by combining optogenetic stimulation of VP GABA or Glut neurons with fiber photometry recordings of DA, GABA or Glut neurons in VTA. We also compared the capacity of VP cell types to evoke a response in VTA when the stimulation was delivered passively by the investigator versus behavior-contingent delivery controlled by the subject. Together these experiments revealed that i) while driving activity in VP GABA and Glut neurons led to opposing behavioral effects, ii) both populations were similarly activated by appetitive approach, novelty, or aversive stimuli. Activity in VP cell types led to differential effects on VTA cell types: iii) stimulation of VP GABA neurons inhibited VTA GABA neurons but activated VTA DA and Glut neurons; however, iv) their ability to recruit DA or Glut neuron activity was markedly inhibited by behavioral contingency, suggesting that the subjects' action or the expected outcome of the action reduced DA neuron excitability. On the other hand, v) VP Glut neurons potently recruited VTA GABA neurons, but also increased activity in DA and Glut neurons, despite driving aversion; however, vi) their ability to affect DA release varied by projection target in NAc. VP Glut neuron activity vii) increased DA release in ventromedial NAc shell where DA release has been prior shown to relate to aversion[43,44], but viii) led to decreased DA release in dorsomedial NAc shell. These data suggest a collaborative role for VP GABA and Glut neurons to titrate adaptive behavioral responses to salient stimuli. Dysregulation of this balance may underlie deficits in reward-seeking and threat-avoidance behaviors associated with drug addiction and other neuropsychiatric disorders.

## Results

### VP GABA and glutamate neurons are both activated by approach to appetitive and aversive reinforcers

We expressed GCaMP calcium sensors in a Cre-dependent manner in the VP of VGAT-Cre or VGLUT2-Cre animals and implanted an optic fiber for detecting bulk calcium activity in VP cell types (Table S1). At the conclusion of experiments all mice were assessed histologically to validate the expression of reporters, opsins, and fiber tract placements (Fig. S1).

We first measured GCaMP responses when mice approached and interacted with (i.e., sniffed, grabbed, or consumed) appetitive high-fat high-sucrose food (HFHS) across two 20-min sessions. Both VP cell types increased activity during HFHS interaction events (0 to 1 s) vs. baseline (−2 to −1s), with signal ramping up as the animal approached HFHS (pre-event, −1 to 0 s) (first 10 HFHS interactions; RM one-way ANOVA; effect of period; GABA cells, $F_{(2,5)} = 19.1$, $p = 0.005$; Glut cells, $F_{(2,6)} = 41.7$, $p < 0.0001$) (Fig. 1A, B). We compared the signals during the first 10 and last 10 HFHS interactions and observed no significant differences, suggesting that both VP cell type stably encode appetitive behavior (event z-score peak; t-test; GABA cells, $t_{(5)} = 0.13$, ns; Glut cells, $t_{(6)} = 0.64$, ns). We also evaluated the response of VP cell types to a non-appetitive novel object, a marble, across two 20-min sessions (Fig. 1C, D). Similar to HFHS, VP GABA and Glut neurons increased activity during the first 10 interactions with the marble, with a ramp-up as the animal approached (first 10 marble interactions; RM one-way ANOVA; effect of period; GABA cells, $F_{(2,5)} = 27.6$, $p = 0.002$; Glut cells, $F_{(2,5)} = 12.0$, $p = 0.01$). However, activity in both VP cell type was less

during the last 10 interactions compared to the first 10 (event z-score peak; t-test; GABA cells, $t_{(5)} = 3.09$, $p = 0.03$; Glut cells, $t_{(5)} = 5.0$, $p = 0.004$). Because increases in activity persisted with approach to HFHS, but waned with approaches to a neutral novel object, these data suggest that activity in VP cell types encode appetitive value as well as novelty, or salience.

To attain more precise temporal resolution for event onsets, we also monitored activity during consummatory licking behavior. Mice freely consumed strawberry milk, spontaneously licking in bouts that can be sorted by variable length inter-lick intervals (ILI). Consistent with HFHS consumption, both VP cell types increased activity at lick initiation, with a ramp-up prior to the detection of the first lick in a bout (ILI 20+s; RM one-way ANOVA; effect of period; GABA cells, $F_{(2,4)} = 19.0$, $p = 0.0009$; Glut cells, $F_{(2,3)} = 20.5$, $p = 0.01$) (Fig. 1E, F). Defining bouts with longer ILIs led to larger increases in activity of both VP cell types (event z-score peak; RM one-way ANOVA; effect of ILI; GABA cells, $F_{(4,4)} = 16.33$, $p = 0.005$; Glut cells, $F_{(4,3)} = 6.4$, $p = 0.04$; Pearson correlation; GABA, $R^2 = 0.6$, $F_{(1,23)} = 34.8$, $p < 0.0001$; Glut cells, $R^2 = 0.59$, $F_{(1,18)} = 26.01$, $p < 0.0001$), suggesting that VP activity is preferentially responsive to the initial approach to an appetitive stimulus and sensitive to the interval between consummatory bouts.

We next investigated VP cell-type responses to a negative-valence food stimulus. We variably provided standard chocolate-flavored 20-mg food pellets or the same pellets coated with quinine, a bitter tastant. Mice ingested all provided chocolate food pellets; they also ate the first few quinine-coated pellets but, by the end of the session, would often reject them (Fig. S2A, B). Activity in both VP cell types increased during interaction with chocolate as well as quinine-coated pellets, with a ramp-up as the animal approached the pellets (chocolate pellets; RM one-way ANOVA; effect of period; GABA cells, $F_{(2,5)} = 13.4$, p = 0.01; Glut cells, $F_{(2,5)} = 10.9$, $p = 0.02$. Quinine pellets; RM one-way ANOVA; effect of period; GABA cells, $F_{(2,5)} = 4.4$, $p = 0.06$; Glut cells, $F_{(2,5)} = 20.8$, $p = 0.005$) (Fig. 2A, B). No clear correlations were observed between the amplitude of calcium signal and type of interaction event or time (Fig. S2). However, VP GABA neuron responses to quinine-coated pellets were significantly less than to chocolate pellets (Fig. 2A), while VP Glut neuron responses to chocolate and quinine-coated pellets were not significantly different (Fig. 2B) (event z-score peak; paired t-test; GABA cells, $t_{(5)} = 4.6$, p = 0.006; Glut cells, $t_{(5)} = 0.88$, p = 0.4). This experiment suggests that VP GABA neurons preferentially respond to palatable taste stimuli, while VP Glut neurons may be less discriminating in this regard.

We then investigated the response of VP cell types to a non-gustatory aversive stimulus. Mice received a brief electric shock (unconditioned stimulus, US) following the conclusion of a 5-s auditory cue (conditioned stimulus, CS). Activity in both VP cell types increased in response to the US (Fig. 2C, D). However, only VP GABA cells responded to the CS, and this response persisted through an extinction session where the CS was presented in the same context but no US was delivered (RM 2-way ANOVA; GABA cells; effect of session, $F_{(1, 4)} = 12.2$, p = 0.02; interaction session x stimulus, $F_{(1, 4)} = 24.0$, p = 0.008; Glut cells; effect of session, $F_{(1, 3)} = 68.2$, p = 0.004; interaction session x stimulus, $F_{(1, 3)} = 13.3$, p = 0.04). VP Glut cells did not respond to the CS during conditioning or extinction (Fig. 2D). These findings suggest that activity in VP GABA but not Glut neurons encodes cues predictive of an aversive stimulus.

### VP GABA neuron stimulation inhibits VTA GABA neurons but activates VTA DA and glutamate neurons

Despite notable differences between VP cell types, the data above show that positive and negative reinforcers induced similar responses in VP GABA and Glut neurons, with both responding during the approach to or delivery of appetitive, novel, or aversive stimuli. To test how similar patterns of activity in two different cell types with purportedly opposite functional roles might lead to opponent behavioral

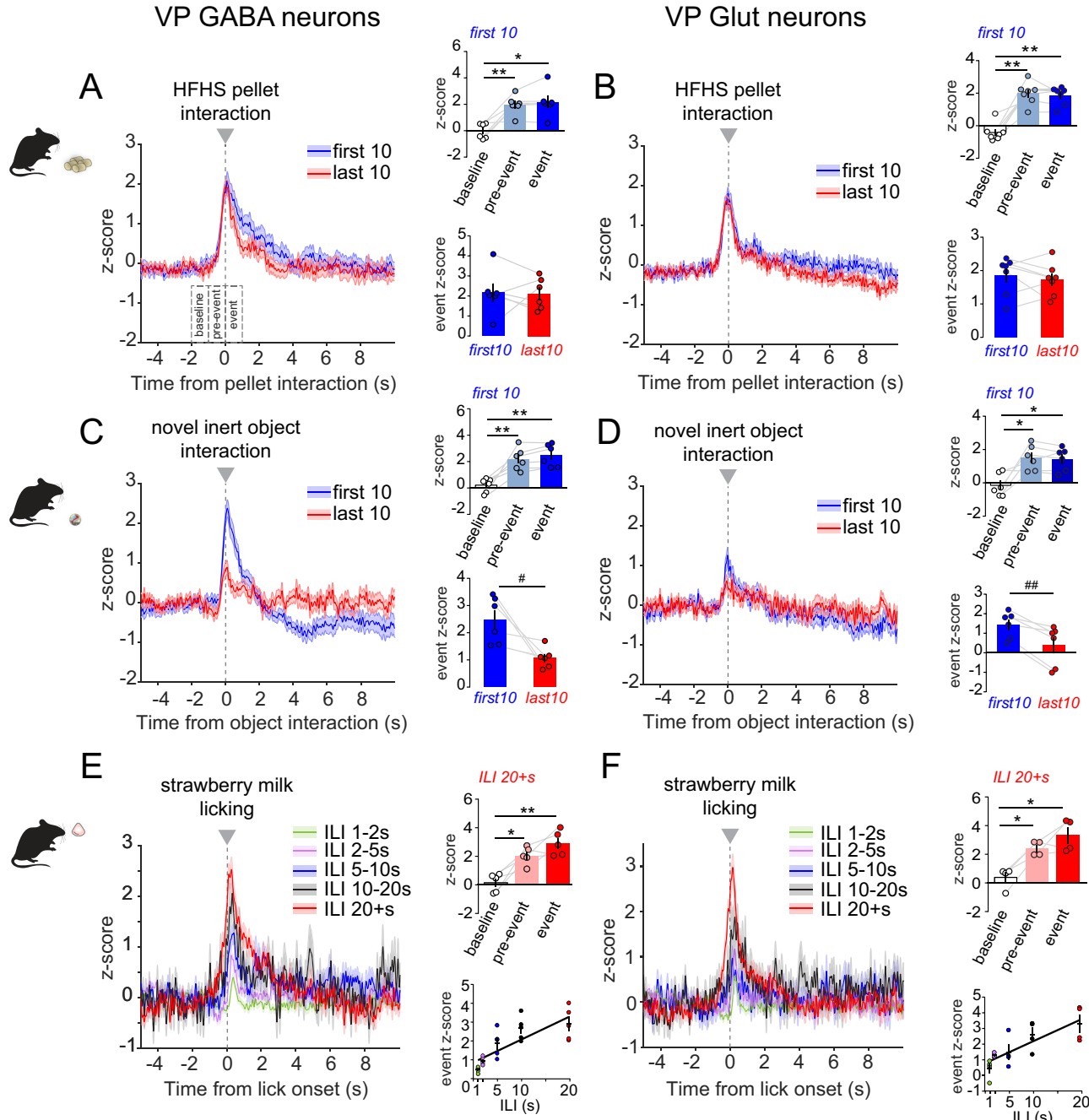

**Fig. 1 | VP GABA and glutamate neurons are activated by approach to rewarding and novel stimuli.** GCaMP fluorescence recorded from **A** VP GABA ($n = 6$ mice) and **B** VP Glut neurons ($n = 7$ mice) upon interaction with a high fat high sucrose (HFHS) food pellet ($t = 0$ s), comparing the first 10 (blue traces) and last 10 interactions (red traces). Insets show mean peak z-score per animal during specified periods (baseline, pre-event, and event) associated with the first 10 HFHS interactions (top), and during the initial 1-s of the first 10 vs. last 10 interactions with HFHS (bottom). GCaMP signals from (**C**). VP GABA ($n = 6$ mice) and (**D**). Glut neurons ($n = 6$ mice) upon interaction with a novel object (marble, $t = 0$ s). Insets similar to A & B. GCaMP signals from (**E**). VP GABA ($n = 5$ mice) and (**F**). Glut neurons ($n = 4$ mice) upon initiating a bout of licking strawberry milk ($t = 0$ s). Licking bouts with 1-

2 s (green), 2-5 s (magenta), 5-10 s (blue), 10-20 s (black), and 20 s or more (red) inter-lick interval (ILI) are represented. Bouts with longer ILI resulted in progressively greater peak GCaMP responses upon bout initiation. Top inset shows mean peak z-score per animal during specified periods (baseline, pre-event, and event) associated with the 20+s ILI bout onset (top). Bottom inset shows mean peak z-score per animal during the initial 1 s of bouts with specified ILIs (1-2, 2-5, 5-10, 10-20 and 20+s), and associated mean linear regression. *$p < 0.05$, **$p < 0.01$ by Tukey post-hoc following one-way ANOVA; #$p < 0.05$, ##$p < 0.01$ by two-tailed paired t-test. Data are represented as mean ± SEM. See Tables S1 and S2, also Fig. S1. Source data are provided as a Source Data file.

responses, we began a series of functional connectivity experiments. We first performed ex vivo experiments using cell-attached and whole-cell electrophysiological recordings from VTA neurons, a major target of both VP cell types, while optogenetically stimulating VP GABA or Glut terminals (Fig. S3A). Following recordings, VTA neurons were

identified as DA or non-DA by TH immunostaining (Fig. S3B). These data show that activation of VP GABA neurons evoked inhibitory postsynaptic currents (IPSC) and inhibited spontaneous firing in both DA and non-DA neurons of VTA (Fig. S3C). Reciprocally, VP Glut neuron activation evoked excitatory postsynaptic currents (EPSC) and

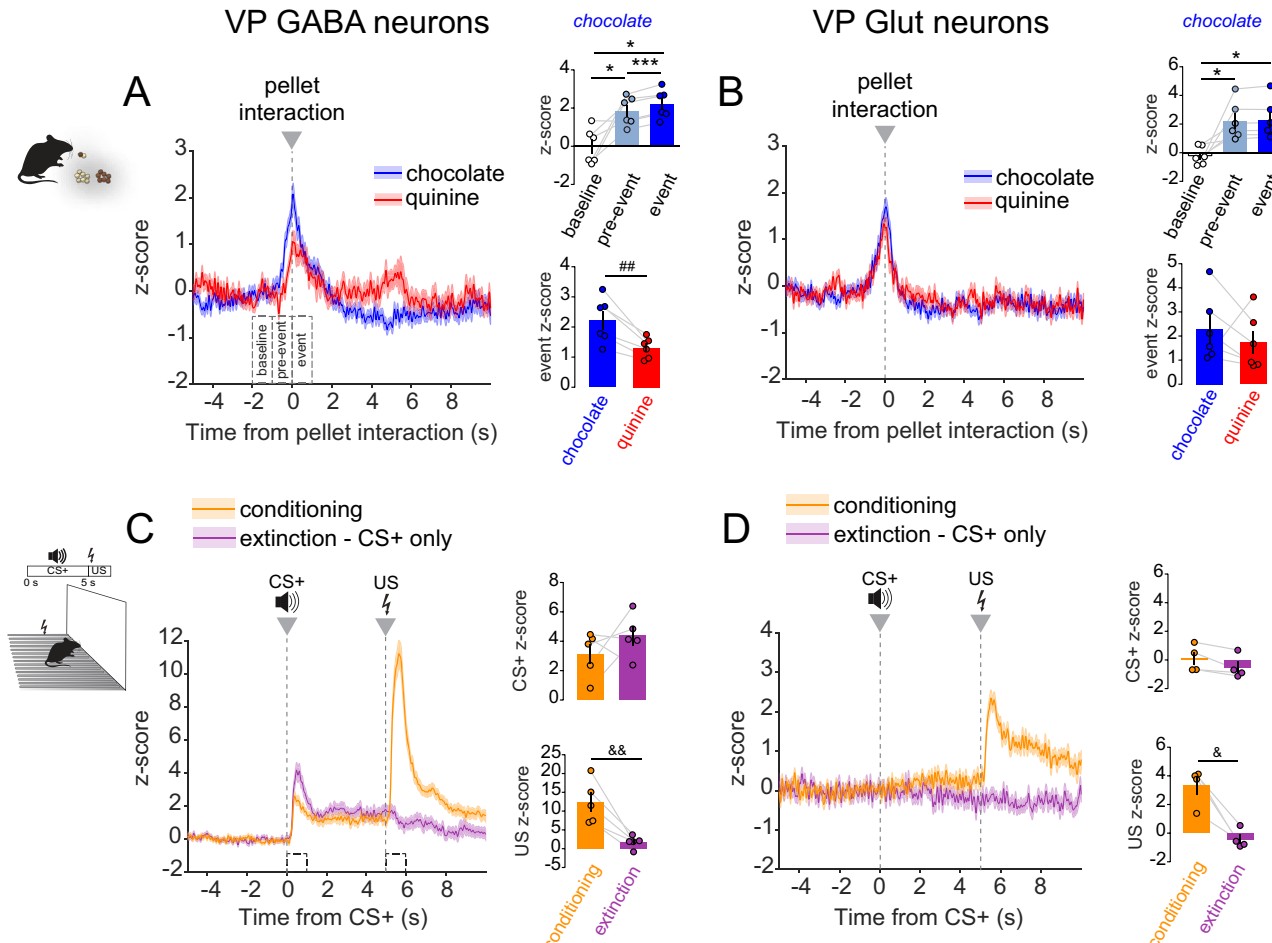

**Fig. 2 | VP GABA and glutamate neurons are activated by aversive stimuli.**
GCaMP fluorescence recorded from (**A**). VP GABA ($n = 6$ mice) and (**B**). VP Glut neurons ($n = 6$ mice) upon interaction with chocolate-flavored (blue traces) and quinine-coated pellets (red traces) (t = 0 s). Insets show mean peak z-score per animal during specified periods (baseline, pre-event, and event) associated with chocolate pellet interaction (top), and during the initial 1-s of interaction with specified pellet type (chocolate and quinine-coated) (bottom). **C** VP GABA ($n = 5$ mice) and (**D**). Glut neuron ($n = 4$ mice) responses upon delivery of a 5-s auditory cue (t = 0 s; conditioned stimulus, CS+) followed immediately by a 0.5 s 0.6 mA foot shock (t = 5 s; unconditioned stimulus, US) during conditioning session (orange traces) and a subsequent extinction session (purple traces) when no shocks were delivered (CS+ only). Insets show mean peak z-score per animal during a 1-s period at the onset of specified stimuli (CS+, top; US, bottom) during specified sessions (conditioning and extinction). $*p < 0.05$, $**p < 0.01$, $***p < 0.001$ by Tukey post-hoc following one-way ANOVA; $^{\#}p < 0.05$, $^{\#\#}p < 0.01$ by two-tailed paired t-test; $^{\&}p < 0.05$, $^{\&\&}p < 0.01$ by Sidak post-hoc following RM two-way ANOVA. Data are represented as mean ± SEM. See Tables S1 and S2, also Fig. S1 and S2. Source data are provided as a Source Data file.

increased spontaneous firing in both DA and non-DA neurons of VTA (Fig. S3D). While VP GABA and Glut neurons may preferentially influence one or another VTA cell type, this approach revealed considerable variability and no clear or significant differences. Moreover, this ex vivo approach has limited power to differentiate how activity in VP GABA or Glut neurons influences VTA cell types at the integrated circuit level in vivo. We thus decided to conduct a series of in vivo experiments using optogenetic stimulation of VP while measuring calcium activity in VTA.

To accomplish this, we used Cre- or FLP-dependent Adeno-associated virus (AAV) vectors to achieve expression of Channelrhodopsin-2 (ChR2) in VP cell types and GCaMP in VTA cell types (Table S1, Fig. S1). We stimulated VP GABA neurons and recorded fluorescence in genetically defined DA, GABA, or Glut neurons in VTA (Fig. 3A). We compared responses to passive stimulation (non-behavior-contingent) in a home-cage environment versus behavior-contingent stimulation using a real-time place preference procedure (RTPP). In RTPP, mice were allowed to traverse between 2 equal-sized compartments where entry into the side designated 'active' initiated 40-Hz stimulation of VP GABA neurons, and exit from the active side terminated stimulation. As

previously observed[33], mice showed a strong preference for the active side and switched side preference when the side designated as active was reversed, indicating that VP GABA neuron stimulation was rewarding (RM 2-way ANOVA, effect of stimulation, $F_{(6, 90)} = 389.4$, $p < 0.0001$) (Fig. 3B).

For passive stimulation we used 40-Hz trains delivered for 5 s (Fig. 3C) because 40 Hz led to the most robust behavioral responses on reinforcement assays[33] and is concordant with phasic activity monitored in vivo[35]. However, we also compared a range of frequencies, durations, and inter-stimulation intervals (ISI) (Fig. S4), observing qualitatively similar effects whose magnitude varied with frequency or duration.

VP GABA neuron stimulation led to a sharp increase in VTA DA neuron activity that peaked $0.45 \pm 0.03$ s after onset, then partially declined before rising, plateauing, then decaying with stimulation offset (z-score peak; RM one-way ANOVA; effect of period; $F_{(2,4)} = 41.4$, $p = 0.001$) (Fig. 3D). A similar pattern emerged when recordings were made from VTA Glut neurons with peak activity occurring $0.48 \pm 0.04$ s after stimulation onset (z-score peak; RM one-way ANOVA; effect of period; $F_{(2,7)} = 27.8$, $p = 0.0001$) (Fig. 3E), consistent with VTA DA and

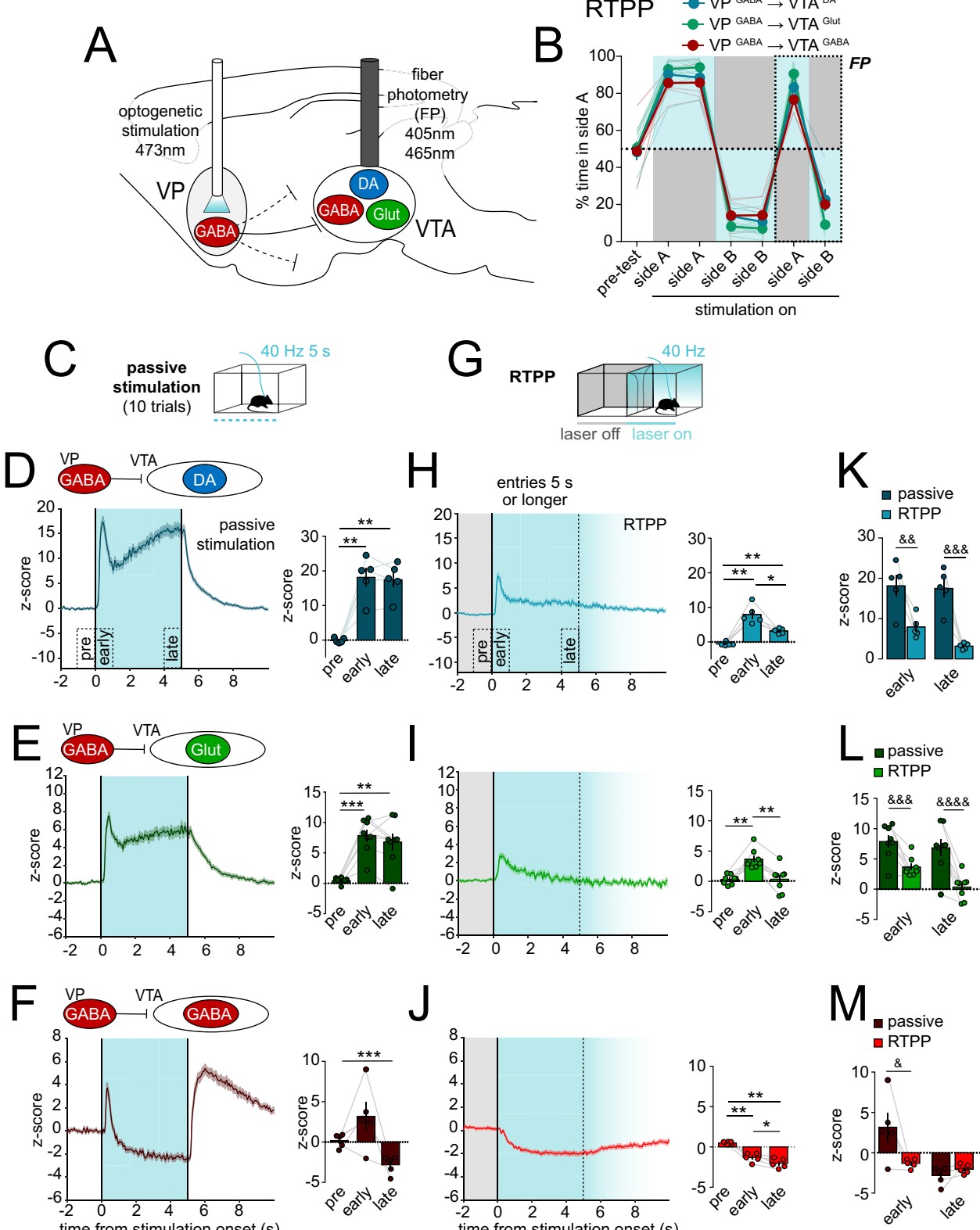

Glut neurons encompassing overlapping populations and roles in positive reinforcement[45–47]. On the other hand, passive stimulation of VP GABA neurons evoked a very different response in VTA GABA neurons – a sharp transient increase that peaked at $0.32 \pm 0.01$ s, followed by a sharp and sustained inhibition below baseline, and a large post-stimulation rebound activation (z-score peak; RM one-way ANOVA; effect of period; $F_{(2,4)} = 9.4$, $p = 0.04$) (Fig. 3F, Fig. S5).

During RTPP, all animals showed a strong preference for the stimulation side throughout the 6-day procedure; VTA activity was recorded only during the final two days (Fig. 3B, G). Events during which the animals spent at least 5 s in the paired side, and were preceded by a period of at least 2 s in the unpaired side, were extracted and averaged (Fig. 3H–J). Both VTA DA and Glut neurons showed a sharp transient increase in activity (z-score peak VTA DA neurons; RM

**Fig. 3 | Optogenetic stimulation of VP GABA neurons drives preference, inhibits VTA GABA neurons, and disinhibits VTA DA and Glut neurons. A** Design of experiment with optogenetic stimulation of VP GABA neurons and simultaneous recording of GCaMP fluorescence in VTA DA ($n = 5$ mice, blue), Glut ($n = 8$ mice, green), or GABA neurons ($n = 5$ mice, red). **B** Time spent in side A during real-time place preference (RTPP) assay with blue shading representing the stimulated side; fiber photometry (FP) recordings were made during the last two sessions. **C** Schematic of the passive stimulation assay. **D** Recordings from VTA DA **E** VTA Glut, or **F** VTA GABA neurons in response to 5-s passive stimulation ($t = 0$ s). Histograms show mean peak z-score per animal during specified 1-s periods (pre-, early- and late-stimulation). **G** Schematic of the RTPP assay; only entries sustained for ≥5 s in the stimulation side (blue shading, fades after 5 s) and preceded by ≥2 s in the no-stimulation side (gray shading) were included. **H**. Recordings from VTA DA **I**. VTA Glut or **J**. VTA GABA neurons in response to stimulation triggered by entry into the active side ($t = 0$). Histograms show mean peak z-score per animal during specified 1-s periods (pre-, early- and late-stimulation). **K–M**. Within-subject comparisons of VTA neuron responses evoked by passive stimulation versus RTPP during early- and late-stimulation. *$p < 0.05$, **$p < 0.01$, ***$p < 0.001$ by Tukey post-hoc following one-way ANOVA; &$p < 0.05$, &&$p < 0.01$, &&&$p < 0.001$, &&&&$p < 0.0001$ by Sidak post-hoc following RM two-way ANOVA. Data are represented as mean ± SEM. See also Tables S1 and S2, also Fig. S1, S3 through S5. Source data are provided as a Source Data file.

one-way ANOVA; effect of period; $F_{(2,4)} = 31.5$, $p = 0.003$. z-score peak VTA Glut neurons; RM one-way ANOVA; effect of period; $F_{(2,7)} = 17.4$, $p = 0.0003$). However, the increase was much smaller than that observed for passive stimulation (z-score peak VTA DA neurons; RM two-way ANOVA; effect of assay; $F_{(1,4)} = 30.5$, $p = 0.005$. z-score peak VTA Glut neurons; RM two-way ANOVA; effect of assay; $F_{(1,7)} = 38.3$, $p = 0.0005$) (Fig. 3K, L). Furthermore, the stimulation-induced increase resulting from entrance into the active compartment was not sustained and instead returned near baseline prior to stimulation offset.

For VTA GABA neurons, VP GABA neuron stimulation during RTPP led to inhibition in activity (z-score peak; RM one-way ANOVA; effect of period; $F_{(2,4)} = 65.0$, $p < 0.0001$) (Fig. 3J), similar to that observed during passive stimulation, but without being preceded by the initial transient increase (z-score peak; RM two-way ANOVA; interaction period x assay; $F_{(1,4)} = 10.2$, $p = 0.03$) (Fig. 3F, M). A rebound activation (overshoot above baseline) was observed at stimulation offset similar to that observed following termination of passive stimulation (Fig. S5). Together these results suggest that activating VP GABA neurons inhibits VTA GABA neurons, but recruits activity in VTA DA and Glut neurons, presumably through a disinhibitory motif. Further, the recruitment of VTA DA and Glut neurons is less pronounced when the result is an expected consequence of the subjects' own action (i.e., volitional entry into the active chamber).

In the RTPP there is a mixture of operant (chamber entries) and contextual (residing in the active chamber) associations that may minimize the precision with which stimulation onset is predicted by the subject. Also, analysis of RTPP data is limited by the relatively few events (chamber entries) and variable durations of stimulation per event. We therefore used an intra-cranial self-stimulation (ICSS) task to compare responses to a) 1-s passive stimulation, b) 1-s stimulation contingent on a behavioral (nosepoke) response with a 20-s timeout period, or c), 1-s stimulation delivered by passive playback (PPB), a replay of the pattern delivered by ICSS for each subject but independent of any nosepoke responses made (Fig. 4A). PPB also allowed us to measure activity during 'futile' nosepokes, i.e., nosepokes that occurred during PPB but did not trigger stimulation. All mice displayed ICSS for VP GABA neuron stimulation with a strong preference for the active over inactive head port (Fig. S6A, B). Note that we also compared responses when using various time-out periods following an active nosepoke in ICSS (Fig. S6C–E).

As before, passive stimulation of VP GABA neurons led to a sharp increase in VTA DA neuron activity, and this response was attenuated for ICSS compared to passive stimulation or PPB (stimulation z-score peak; RM one-way ANOVA; effect of assay; $F_{(2,4)} = 28.9$, $p = 0.0009$) (Fig. 4B); again suggesting that VP GABA stimulation when evoked by the subjects' action reduced the resulting VTA DA neuron response. We observed no pre-stimulation ramp in DA neuron activity in the ICSS assay, nor during futile nosepokes in PPB (pre-event z-score peak; RM one-way ANOVA; effect of assay; $F_{(3,4)} = 2.3$, $p = 0.2$) (Fig. 4C), suggesting DA neurons did not in themselves encode the action of reward seeking or nosepoke initiation in this assay.

Glut neurons in VTA were also recruited by activation of VP GABA neurons across all stimulation assays, and displayed more robust activation during passive stimulation and PPB compared to ICSS (stimulation z-score peak; RM one-way ANOVA; effect of assay; $F_{(2,7)} = 9.6$, $p = 0.009$) (Fig. 4D). Thus, similar to the RTPP, VTA DA and Glut neurons were activated by VP GABA neurons. Unlike DA neurons, VTA Glut neurons appeared to display a subtle activity ramp prior to stimulation onset in ICSS, and during futile nosepokes (pre-event z-score; RM one-way ANOVA; effect of assay; $F_{(3,7)} = 3.9$, $p = 0.06$) (Fig. 4E).

VP GABA neuron ICSS caused transient VTA GABA neuron activation, followed by inhibition below baseline, then rebound excitation (Fig. 4F, Fig. S5). Unlike VTA DA or Glut, VTA GABA neuron responses were similar whether stimulation was delivered by ICSS or passive methods (stimulation z-score peak; RM two-way ANOVA; effect of assay; $F_{(2,8)} = 0.004$, $p = 0.99$). However, we detected a pronounced ramping in VTA GABA neuron activity prior to nosepoke in ICSS, as well as during futile nosepokes during PPB (pre-event z-score peak; RM one-way ANOVA; effect of assay; $F_{(3,4)} = 16.6$, $p = 0.0016$) (Fig. 4G). These data suggest that VTA GABA neurons are transiently activated during the initiation of a nosepoke, and that activity in VTA GABA neurons in advance of a pursued reinforcer (here the reinforcer is VP GABA neuron stimulation) may inhibit the resulting DA and Glut neuron responses through local release of GABA onto these cells[48].

## VP glutamate neurons activate VTA DA, glutamate, and GABA neurons

Using the same strategy as in Fig. 3, we next tested the effects of VP Glut neuron stimulation while measuring responses in VTA DA, GABA or Glut neurons to passive stimulation and in the RTPP (Fig. 5A). We and others have prior shown that, in contrast to the reinforcing effects of VP GABA neurons, VP Glut neuron stimulation is aversive and leads to behavioral avoidance[33–35], an effect we reproduced here in all cohorts (RM 2-way ANOVA, effect of stimulation, $F_{(6, 96)} = 265.6$, $p < 0.0001$) (Fig. 5B).

We first assessed responses to passive stimulation (40 Hz, 1 s and 5 s trains) (Fig. 5C) and found that VP Glut neuron stimulation increased net activity in each of VTA DA (1 s stimulation; *t-test*, $t_{(5)} = 13.7$, $p < 0.0001$; 5 s stimulation; RM one-way ANOVA, effect of period, $F_{(2,5)} = 42.7$, $p = 0.0001$) (Fig. 5D), Glut (1s-stimulation; *t-test*, $t_{(6)} = 7.8$, $p < 0.0001$; 5 s stimulation; RM one-way ANOVA, effect of period, $F_{(2,6)} = 50.5$, $p < 0.0001$) (Fig. 5E), or GABA neurons (1 s stimulation; *t-test*, $t_{(5)} = 5.9$, $p = 0.002$; 5 s stimulation; RM one-way ANOVA, effect of period, $F_{(2,5)} = 73.9$, $p = 0.0002$) (Fig. 5F). Different frequencies (Fig. S7A–C), durations (Fig. S7D–F), and ISIs (Fig. S7G–I) showed qualitatively similar responses whose magnitude varied by frequency; with the notable disparity that low-frequency stimulation (5 or 10 Hz) led to suppression rather than recruitment of VTA DA neuron activity (Fig. S7A). We aimed to compare these responses to those evoked in RTPP, however owing to the aversive consequences of VP Glut neuron stimulation, mice made few entries into the active side that were sustained for 5 s (Fig. S8). We thus focused our analysis on all entries that persisted for at least 1 s in the active compartment and were preceded by at least 2 s outside of the active compartment. Similar to passive stimulation, behavior-contingent stimulation in RTPP led to increased VTA DA (*t-test*, $t_{(5)} = 3.8$, $p = 0.01$) (Fig. 5G), Glut

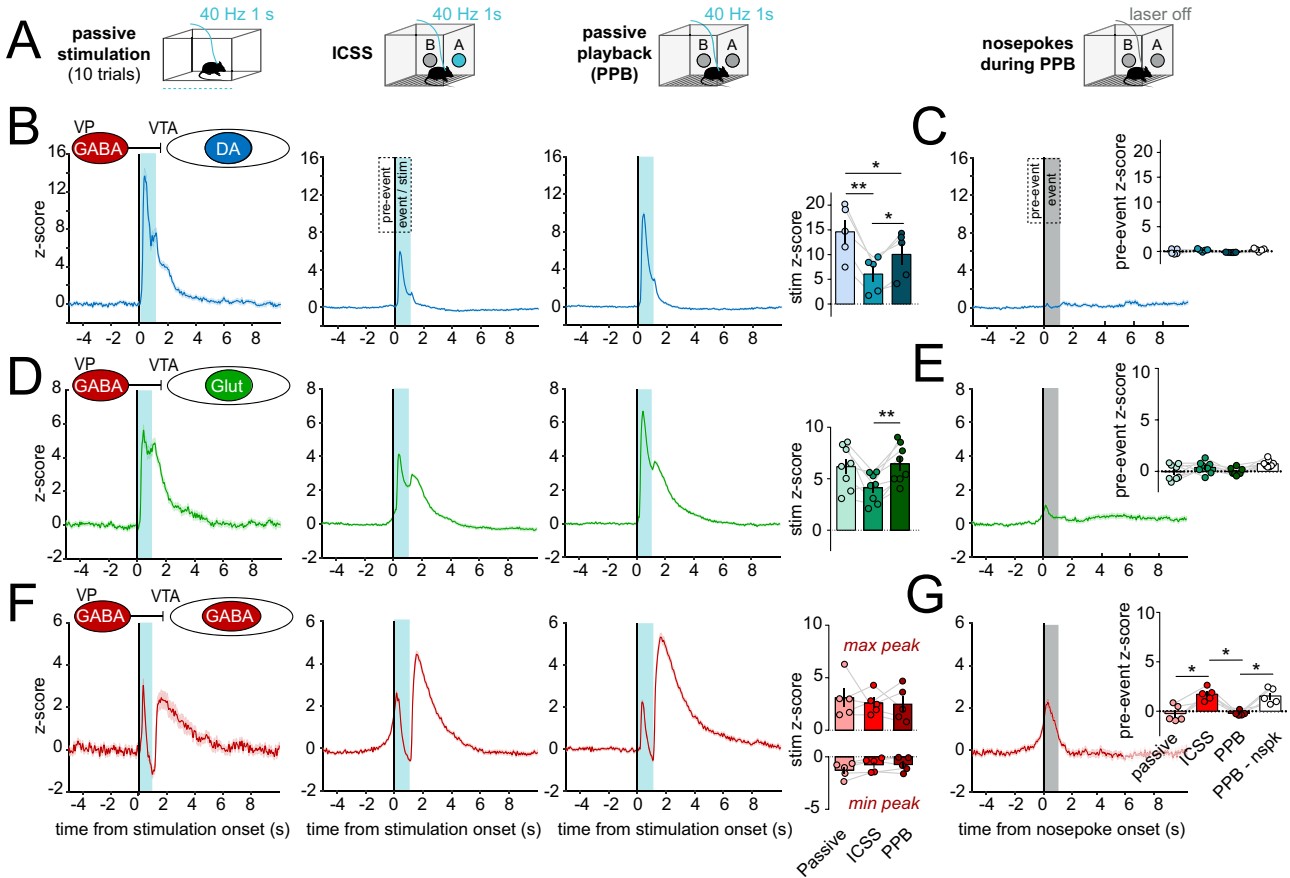

**Fig. 4 | Behavioral contingency modulates VTA responses to VP GABA neuron stimulation. A** Schematic illustrating passive, intracranial self-stimulation (ICSS), and passive playback (PPB, replay) modes of VP GABA neuron stimulation above corresponding datasets. All stimulations were delivered with ≥20-s inter-stimulation interval (ISI). **B** GCaMP recordings from VTA DA neurons during passive, ICSS, or PPB stimulation of VP GABA neurons, from left to right (t = 0 s, 1-s stimulation shaded in blue). Histograms show the mean peak z-score per animal during the 1-s stimulation (event) of specified assays. **C** VTA DA neuron responses to futile nosepokes (t = 0 s) during PPB that did not trigger stimulation (gray shading, PPB - nspk). Histogram shows mean peak z-score per animal during the 1-s pre-event period of specified assays (n = 5 mice, blue). **D, E** VTA Glut neuron responses (n = 8 mice, green). **F, G** VTA GABA neuron responses (n = 5 mice, red). *p < 0.05, **p < 0.01 by Tukey post-hoc following one-way ANOVA. Data are represented as mean ± SEM. See Tables S1 and S2, also Fig. S1, S3 through S6. Source data are provided as a Source Data file.

(t-test, $t_{(6)}$ = 8.4, p = 0.0002) (Fig. 5H), and GABA neuron activity (t-test, $t_{(5)}$ = 4.7, p = 0.006) (Fig. 5I).

### VP GABA and glutamate neurons differentially evoke versus suppress DA release in dorsomedial shell

Our cell-type-specific VTA recordings are defined by neurotransmitter content, however, even within transmitter-defined VTA neurons there is molecular, anatomical, and functional diversity[49–52] that is blended in bulk fiber photometry recordings. Indeed, this diversity may explain how both VP GABA and VP Glut neuron stimulation can lead to a net increase in bulk DA neuron activity despite driving opposing effects on behavior, that is, if VP GABA and Glut neurons recruit different populations of DA neurons in VTA. Indeed, while DA neuron activity is highly sensitive to rewards, subsets of DA neurons are recruited by aversive stimuli[53–57], and recent evidence suggests that DA release in the ventromedial shell of the NAc (vmsh) is associated with aversion while DA release in dorsomedial NAc shell (dmsh) is associated with reward[44].

To test if VP GABA versus Glut neurons differentially drive DA release in dmsh and vmsh we turned to the DA sensor dLight[58]. We expressed dLight in NAc and fibers were implanted to measure DA release in the two subregions. In addition, ChR2 was expressed in VP GABA or Glut neurons with fibers implanted in VTA to activate VP projections to VTA (Fig. S1). We then assessed DA signals in response to

stimulation (Fig. 6, Fig. S9, S10) using the same assays as described in preceding experiments.

VP GABA neuron stimulation evoked DA release in both dmsh (Fig. 6A) and vmsh (Fig. 6B), and this occurred whether stimulation was delivered via ICSS, PPB, RTPP, or passively. Moreover, the patterns of DA release measured using dLight in NAc were highly concordant with the patterns of calcium activity measured using GCaMP in VTA. For example, PPB evoked more DA release than did behavior-contingent stimulation by ICSS (stimulation peak z-score NAc dmsh; t-test, $t_{(5)}$ = 5.2, p = 0.003. stimulation peak z-score NAc vmsh; t-test, $t_{(4)}$ = 3.9, p = 0.02). Further, DA release was sustained in response to prolonged (5 s) passive stimulation but peaked and then decayed when stimulation was behavior-contingent in the RTPP (peak z-score NAc dmsh; RM two-way ANOVA, interaction period x assay, $F_{(1,5)}$ = 31.7, p = 0.002. peak z-score NAc vmsh; RM two-way ANOVA, interaction period x assay, $F_{(1,5)}$ = 43.3, p = 0.001). These data support the conclusion that when the reinforcer was an expected consequence of the subject's action, the evoked DA response was smaller.

Finally, we found that stimulation of VP Glut projections to VTA led to opposing patterns of DA release in NAc subregions, suppressing DA release in dmsh (stimulation peak z-score NAc dmsh; 1-s passive stimulation, t-test, $t_{(4)}$ = 2.71, p = 0.03; RTPP, t-test, t(4) = 2.13, p = 0.049; 5-s passive stimulation, RM one-way ANOVA, period effect, $F_{(4,8)}$ = 12.62, p = 0.02) (Fig. 6C), but evoking DA release in vmsh

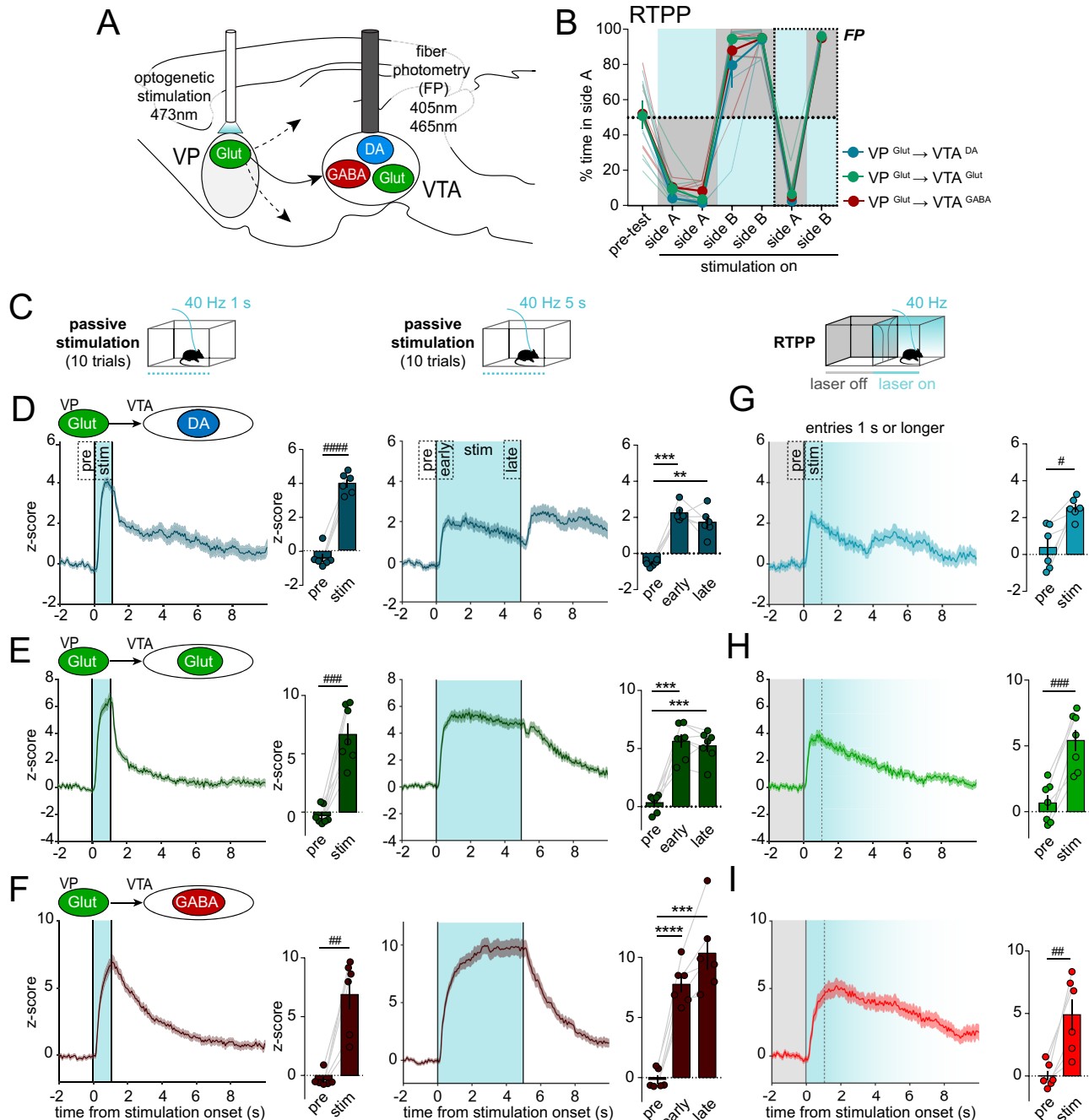

**Fig. 5 | Optogenetic stimulation of VP Glut neurons drives avoidance, activates VTA GABA, DA and Glut neurons. A** Design of experiment with optostimulation of VP Glut neurons and simultaneous recording of GCaMP fluorescence in VTA DA ($n = 6$ mice, blue), Glut ($n = 7$ mice, green), or GABA neurons ($n = 6$ mice, red). **B** Time spent in side A during real-time place preference (RTPP) assay with blue shading representing stimulated side; fiber photometry (FP) recordings were made during the last two sessions. **C** Schematic illustrating passive stimulation and RTPP assays. **D** GCaMP responses of VTA DA, **E** VTA Glut, or **F** VTA GABA neurons during the 1-s or 5-s passive stimulation (shaded in blue, onset at $t = 0$ s). Histograms show mean peak z-score per animal during 1-s specified periods (pre-stimulation and stimulation for the 1-s stimulation; pre-, early- and late-stimulation for the 5-s stimulation assay). **G** GCaMP responses of VTA DA, **H** VTA Glut, or **I** VTA GABA

neurons in response to stimulation induced by entry into the active side during RTPP ($t = 0$ s); only RTPP entries sustained for ≥1 s in the stimulation side (blue shading, fades after 1 s) and preceded by ≥2 s in the no-stimulation side (gray shading) were included. Histograms show mean peak z-score per animal during 1-s specified periods (pre-stimulation and stimulation). Statistics on histograms are results of t-test for D-F (left) & G-I, and of Tukey post-hoc following one-way ANOVAs for D-F (right). *$p < 0.05$, **$p < 0.01$, ***$p < 0.001$, ****$p < 0.0001$ by Tukey post-hoc following one-way ANOVA; #$p < 0.05$, ##$p < 0.01$, ###$p < 0.001$, ####$p < 0.0001$ by two-tailed paired t test. Data are represented as mean ± SEM. See Tables S1 and S2, also Fig. S1, S3, S7 and S8. Source data are provided as a Source Data file.

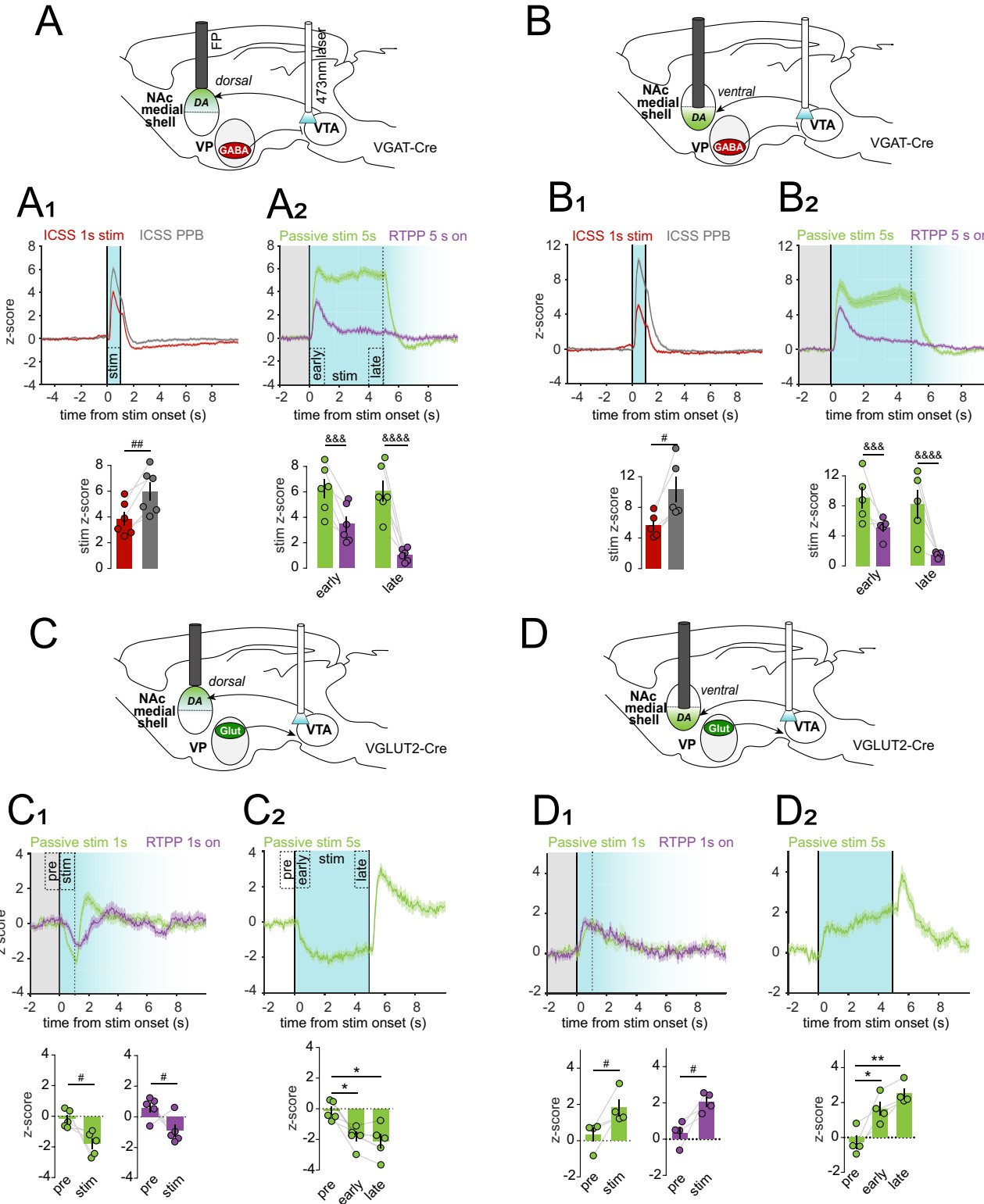

(stimulation peak z-score NAc vmsh; 1-s passive stimulation, *t-test*, $t_{(3)} = 2.7$, $p = 0.04$; RTPP, *t-test*, $t_{(3)} = 3.3$, $p = 0.02$; 5-s passive stimulation, RM one-way ANOVA, period effect, $F_{(2,3)} = 20.1$, $p = 0.002$) (Fig. 6D). Together, these results validate our findings using GCaMP recordings in VTA by using an independent measure of DA release and with pathway-specific stimulation of VP projections to VTA. Moreover, the dLight experiments demonstrate that activation of VP Glut neurons selectively suppresses DA release in reward-associated dorsomedial, but enhances DA release in aversion-associated

ventromedial NAc shell, consistent with a role for VP Glut neurons driving behavioral avoidance.

## Discussion

VP neural activity has been linked to multiple reward-related processes, including motivation, reward expectation, and reward prediction error (RPE). For instance, in vivo electrophysiological studies observed some VP cells that increased while others decreased their activity in response to predictive cues and subsequent reward

**Fig. 6 | Optogenetic stimulation of VP terminals in VTA reveals divergent patterns of DA release in NAc subregions. A** Experimental design for optogenetic stimulation of VP GABA terminals in VTA and simultaneous measurement of dLight in dorsal medial shell of NAc (dmsh) ($n = 6$ mice). **A1** dLight response to behavior-contingent ICSS (red trace) and non-contingent PPB (gray trace) (t = 0 s, stimulation period shaded blue). Histogram shows mean peak z-score per animal during the 1-s stimulation of specified assays (ICSS, red; PPB, gray). **A2** dLight response to behavior-contingent RTPP (purple trace) and non-contingent 5-s passive stimulation (green trace); only RTPP entries sustained for ≥5 s in the stimulation side (blue shading, fades after 5 s) and preceded by ≥2 s in the no-stimulation side (gray shading) were included. Histogram shows mean peak z-score per animal during the early- and late-stimulation periods of specified assays (passive stim, green; RTPP, purple). **B, B1 & B2** As in A, A1 & A2, but with dLight recordings in the ventral medial shell of the NAc (vmsh) ($n = 5$ mice). **C.** Experimental design for optogenetic

stimulation of VP Glut terminals in VTA and simultaneous measurement of dLight in dmsh ($n = 5$ mice). **C1** dLight response to 1-s passive stimulation (green trace) and active side entries in RTPP (purple trace); only entries sustained for ≥1 s in the stimulated side (blue shading, fades after 1 s) and preceded by ≥2 s in the no-stimulation side (gray shading) were included. Histograms shows mean peak z-score per animal during specified periods (pre-stimulation and stimulation) of the passive stim (green) and RTPP (purple) assays. **C2** dLight response to 5-s passive stimulation and associated histogram showing mean peak z-score per animal during specified periods (pre-, early-, and late-stimulation). **D, D1 & D2** As in C, C1 & C2 but with dLight recordings in vmsh ($n = 4$ mice). \*$p < 0.05$, \*\*$p < 0.01$ by Holm-Sidak post-hoc following one-way ANOVA; #$p < 0.05$, ##$p < 0.01$, by paired t-test; &&&$p < 0.001$, &&&&$p < 0.0001$ by Sidak post-hoc following RM two-way ANOVA. Data are represented as mean ± SEM. See Tables S1 and S2, also Fig. S1, S5, S9 and S10. Source data are provided as a Source Data file.

seeking[20]. Moreover, a larger population of VP neurons is modulated by reward delivery, predictive cues, or by both, while a smaller population encode RPE[59,60]. Other work suggests a specific role for VP in impulsive and compulsive behavior[61]. And multiple studies have demonstrated that VP activity relates to and can exert profound control over drug-seeking and other behaviors related to neuropsychiatric disorders[23,25,62–65]. For these reasons, it is of high importance to understand how VP activity shapes activity within downstream circuits.

The heterogeneous firing patterns within VP suggest that distinct VP cell types subserve distinct functional roles. Indeed, recent studies have begun to discriminate how neurochemically defined VP cell types differentially contribute to behavior. Inhibitory GABA-releasing neurons represent the majority population within VP, but excitatory Glut neurons project in parallel and appear to play opponent roles. We and others found that activation of VP GABA neurons supports behavioral reinforcement, while VP Glut neurons elicit avoidance behavior, and their ablation impaired taste aversion learning[33,34,37]. More recently, Stephenson-Jones and colleagues used single-unit recordings to show that VP GABA neurons increased activity in response to water reward but decreased in response to an aversive air puff, while VP Glut neurons responded in an opposite manner[35]. Combined, these results suggest that VP GABA neurons respond to reward and their activity can drive reward seeking, while VP Glut neurons respond to aversive stimuli and drive avoidance behavior.

In the current study we began by using fiber photometry to measure bulk calcium signal from neurochemically defined VP cells in freely behaving mice. VP GABA neurons responded to stimuli that elicit approach, including appetitive food and novel objects; but also responded to aversive stimuli that elicit avoidance. Our results show that VP GABA neurons begin responding prior to the onset of physical interaction with an appetitive stimulus, that activity responses peaked at the time of initial interaction (e.g., initiation of consumption) and that the responses are highly sensitive to the time elapsed since prior interaction. These results appear consistent with the theory that VP cells encode reward value and reward expectation, and suggest that VP cells become more active as subjects initiate a volitional or goal-directed action[20,59,60]. Remarkably, we found that VP Glut neuron responses were very similar to those observed for VP GABA neurons, consistent with a recent report that used similar approaches to assess VP Glut neuron responses[66].

It is, however, unclear why these results using photometric recordings differ from the elegant single-unit work showing that VP GABA cells were activated by appetitive stimuli while VP glutamate neurons were activated by aversive stimuli[35]. One likely possibility is the difference in recording approach. We measured bulk signal from a blend of VP GABA or Glut cells that may themselves be functionally heterogeneous, perhaps including some that would not have met the categorical criteria used by Stephenson-Jones et al. Indeed, that study also identified VP cells that were activated by both appetitive and aversive stimuli, and others that were inhibited by both types of

stimulus, and these cells may be driving the signals we observed. It is also the case that changes in calcium are not always a strong correlate of changes in firing rate. For example, recordings in NAc suggest GCaMP fluorescence and firing rate are poorly correlated, perhaps due to prominent dendritic calcium signals that are better correlates of afferent drive rather than firing[67,68]. Another contributing factor may relate to the differences in task constraints, i.e., VP cells may respond differently in highly trained head-fixed mice compared to untrained mice freely exploring an environment. Indeed, if VP unit responses to appetitive or aversive stimuli vary with changes in state[21], then different modes of responding to such stimuli may predominate in different behavioral contexts or as subjects transition between states.

VP GABA and Glut neurons send projections to multiple brain regions, but one region to which they both project densely is the VTA[3,32–34], and this pathway has been prior studied in the context of behavioral disorders[38,42,62,64]. Behaviors induced by stimulation of VP cell types were recapitulated by stimulation of their VTA terminals, further supporting the importance of the VP to VTA pathway in approach and avoidance behaviors[33]. We hypothesize that VP GABA and glutamate neurons are similarly responsive to rewarding/aversive stimuli but drive different behavioral responses to those stimuli through differential recruitment of VTA cell types. VP could thus modulate approach/avoid behaviors in the face of conflicting motivations, with both pathways functioning either synergistically or in competition. Prior ex vivo works suggest that diverse VTA cell types receive relatively unbiased/similar levels of input from VP[69,70], a result consistent with the slice electrophysiology data included in this manuscript. But because such ex vivo assessments of functional connectivity obscure polysynaptic or state-dependent effects of VP activity in the presence of ongoing network activity, we focused on an in vivo approach to investigate how VP GABA or Glut activity independently modulates activity of VTA cell types during behavior.

Importantly, our experimental approach of stimulating VP cell types while recording from VTA cell types does not demonstrate that the effects of VP on VTA cell types are mediated solely by direct projections between these structures. Indeed, optogenetic manipulation of VP can induce a multitude of effects mediated through local VP collaterals, projections to lateral habenula, lateral hypothalamus, and other regions that also regulate VTA activity. Thus, while VP GABA and glutamate neurons have a prominent projection to VTA where they synapse directly on to diverse cell types, polysynaptic circuit mechanisms through other brain regions are assuredly contributing to the effects we observed on VTA. Yet, that we were able to observe very similar patterns whether stimulating VP cell bodies and recording VTA GCaMP signals, or by stimulating VP terminals in VTA while recording NAc DA release, does imply that direct projections from VP to VTA are sufficient to evoke the observed behavioral and physiological responses.

Overall, we observed that in vivo passive stimulation of VP GABA neurons evoked a large increase in activity in VTA DA, an apparently

smaller increase in VTA Glut, and inhibited VTA GABA neurons; VP Glut neuron stimulation evoked the opposite pattern, a modest increase in VTA DA, an intermediate increase in VTA Glut, and large and sustained activation of VTA GABA neurons (Fig. 7A). When comparing these responses to the ones elicited by behavior-contingent VP GABA stimulation, VTA DA and Glut neurons showed smaller and less sustained responses to VP GABA stimulation (Fig. 7A).

Prior work showed that stimulation of VP Glut neurons led to behavioral avoidance without significant Fos induction in DA neurons[33]. We were thus surprised to find that VP Glut stimulation activated VTA DA neurons along with the other VTA cell types. However, an emerging literature suggests that a subset of DA neurons in VTA is particularly sensitive to aversive stimuli and that activation of these can induce behavioral avoidance[44,53,54]. These aversion-related VTA DA signals appear to emanate from DA neurons in medial VTA, lateral substantia nigra, and more caudal DA sub-groups that project to different target regions, such as medial prefrontal cortex, basolateral amygdala, tail of the striatum, and vmsh NAc[44,54,71,72]. We thus hypothesized that VP Glut neurons may preferentially activate aversion-related DA neurons. To test this we compared the effects of stimulating VP Glut projections in VTA on evoked DA release in two neighboring regions of the medial NAc that were prior shown differentially responsive to aversive footshock[44]. Consistent with prior work and our hypothesis, we found that VP Glut neurons evoked an increase in DA release in aversion-related vmsh NAc, but a decrease in reward-related dmsh NAc. These data are summarized in the proposed circuit model of Fig. 7B.

Moreover, we previously reported that stimulating VP cells below 20 Hz was insufficient to induce preference/avoidance behavior[33]. Here we find that stimulation of VP GABA neurons at 5 or 10 Hz was also insufficient to inhibit VTA GABA neurons and did not support a sustained activation of VTA DA neurons or NAc DA release, consistent with the lack of reward behavior in the RTPP and ICSS assays. Similarly, VP Glut neuron stimulation at lower frequencies did not support an increase in VTA DA neuron activity nor DA release in NAc vmsh, consistent with the lack of avoidance behavior.

Calcium activity ramped in both VP GABA and Glut neurons just prior to the onset of an interaction with rewarding, novel, or aversive stimuli. This suggests that VP GABA and Glut neurons respond to rewarding, novel and salient events, but also encode the subjects' actions in pursuit of a goal or expected reward, as observed in previous recordings from VP[66,73] and VTA[74-76]. Indeed, we also observed an activity ramp in VTA GABA but not DA neurons prior to a nosepoke during ICSS for VP GABA neuron stimulation. This ramp was particularly apparent during passive playback when mice would perform a nosepoke, but no stimulation was delivered. These data are in line with and add to the growing literature suggesting a role for VTA GABA and Glut neurons in reward-seeking or prediction[48,76-80], with activity of these cells ramping up in expectation of reward delivery, while DA neurons report RPE[81-83].

Similar to RPEs in Pavlovian conditioning tasks, DA neuron responses to reinforcing stimuli in operant tasks can be smaller when the reinforcer is the expected result of the subjects' own action[84-86]. Consistent with this, we observed that VTA DA neuron responses to VP GABA neuron stimulation were smaller when delivered in a behavior-contingent manner. This result was observed in multiple behavioral assays, and whether measuring GCaMP signals in VTA or dLight signals in NAc. A similar effect was observed in VTA glutamate neurons, consistent with prior work showing that ~20% of VTA glutamate neurons express DA markers[45,52] and that activity in VTA glutamate projections to NAc can promote behavioral reinforcement[46,47]. Moreover, when VP GABA stimulation was sustained for 5 s, the VTA DA neuron response was biphasic, with a rapid peak followed by a slower ramp, but the slow ramp was absent when stimulation delivery was behavior-contingent. A recent study indicates a similar slow signal in DA neurons is mediated by peptide release from GABA neurons in lateral hypothalamus[87].

Future investigations could explore the presence of a similar peptide mechanism from VP GABA neurons and how such signals relate to reward pursuit or expectation.

On the other hand, VTA GABA neurons were similarly inhibited by VP GABA neurons whether stimulation was delivered by experimenter or self-administered by the subject. In addition, VTA GABA neurons showed a prominent transient increase in activity in response to VP GABA neuron stimulation prior to sustained inhibition. This transient activation began to ramp prior to an operant nosepoke and was present even without stimulation in PPB, demonstrating that it can occur as a consequence of behavior independent of VP stimulation. Yet the transient activation was also present with 'unexpected' passive stimulation, but here initiation was delayed until after stimulation onset. This transient increase in VTA GABA neuron activity was, curiously, absent when delivered in a behavior-contingent manner during the RTPP; perhaps because the action of entering the 'active' chamber is temporally and contextually less precise.

Another remaining question is what circuits drive the transient activation of VTA GABA neurons, both prior to stimulation in ICSS and in response to experimenter-delivered VP GABA neuron activation. This could be mediated by projections from forebrain regions such as cortex, caudal regions including superior colliculus, laterodorsal tegmental area, or any number of other excitatory inputs[69,88-90]. Another possibility is that the transient increase may reflect activity in a subpopulation of VTA GABA neurons that is distinct from those that are inhibited, and that activity in one subtype of GABA neuron drives subsequent inhibition of another subtype. We also noted a prominent rebound activation of VTA GABA neurons, observed as an overshoot above baseline following termination of VP GABA stimulation. DA release in dmsh NAc was inhibited by stimulation of VP glutamate terminals in VTA, and here too, a rebound of DA release was observed. Rebound activation of DA neurons has been observed before and may signal the termination of an aversive stimulus[91-93] or function as a mechanism to reset and restore coherent activity in neuronal ensembles[94-96].

Prior studies support the role of VTA GABA neurons in reward prediction and goal-directed behavior. Steffensen and colleagues[80] showed that putative VTA GABA neurons increased their activity before ICSS of the medial forebrain bundle. Another study showed that optotagged VTA GABA neurons increased their activity during the delay between CS and US[79]. Our data show a ramp in VTA GABA neuron activity prior to a nosepoke in expectation of a reward delivery (during ICSS and PPB), and a decrease in VTA DA and Glut responses when reward was the result of the animal's action. We thus hypothesize that operant actions in pursuit of reward recruit VTA GABA neurons and, through local GABA release, blunt VTA DA and Glut neuron responses to expected rewards. Future work could test the generalizability of this hypothesis to other excitatory and inhibitory inputs or using natural reinforcers.

The results described above suggest that both VP GABA and Glut neurons can become activated in response to salient rewarding, novel, or aversive stimuli but drive markedly different responses in VTA cell types, explaining how their activity can drive opponent approach and avoidance responses. Moreover, the ability of VP to modulate VTA neurons is potently modified by whether the animal is engaged in behaviors that lead to activation of VP, perhaps due to microcircuit mechanisms local to VTA. Dysregulation of VP activity is likely to contribute to maladaptive behaviors associated with compulsive reward-seeking common in drug addiction and other neuropsychiatric disorders.

## Methods
### Animals
VGLUT2-IRES-Cre, VGAT-IRES-Cre, DAT-IRES-Cre, and VGAT-2A-FlpO-D knock-in mice were obtained from The Jackson Laboratory:

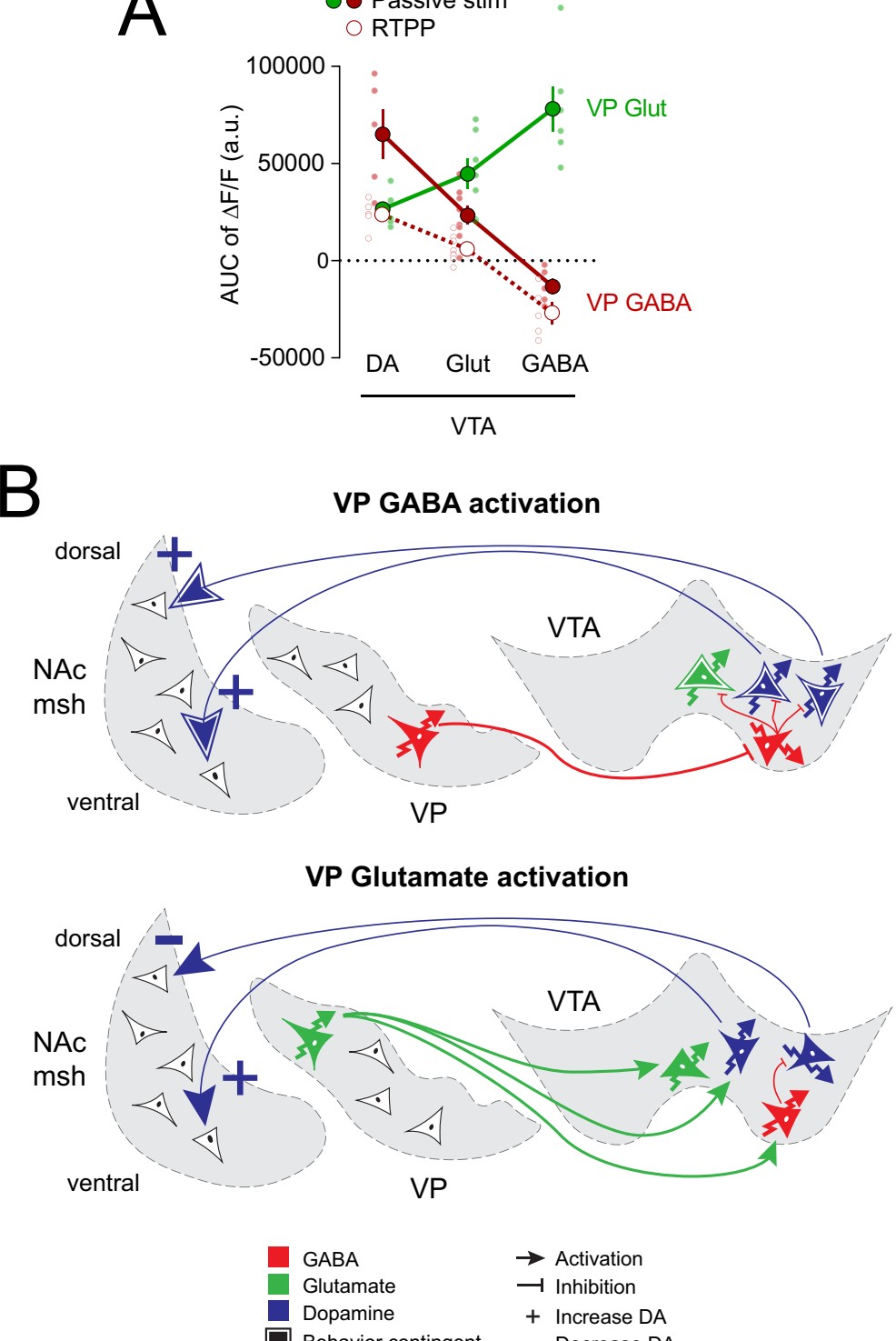

**Fig. 7 | VP GABA and glutamate neurons evoke opposite activity pattern in VTA cell types. A** AUC of ΔF/F (%) was calculated from VTA cell-type responses during 5 s passive stimulation of VP GABA and Glut neurons (5 s, 40 Hz; 20 s ISI) and show opposite response patterns. AUC of ΔF/F from VTA cell-type responses to 5 s VP GABA stimulation during the RTPP assay (including only entries preceded by at least 2 s in the inactive side, followed by at least 5 s in active side) evokes smaller responses in VTA DA and Glut neurons ($n = 5$ mice for VP GABA − VTA DA and VP GABA − VTA GABA, $n = 6$ mice for VP Glut − VTA DA and VP Glut − VTA GABA, $n = 7$ mice for VP Glut − VTA Glut, and $n = 8$ mice for VP GABA − VTA Glut). Data are represented as mean ± SEM. **B** Functional connectivity model by which VP GABA and glutamate neurons differentially recruit activity within VTA cell types. Source data are provided as a Source Data file.

*Slc17a6^(tm2(cre)Lowl)* (RRID:IMSR_JAX:016963), *Slc32a1^(tm2(cre)Lowl)* (RRID:IMSR_JAX:016962), *Slc6a3^(tm1.1(cre)Bkmn)* (RRID:IMSR_JAX:006660) and *Slc32a1^(tm1.1(flpo)Hze)* (RRID:IMSR_JAX:029591). The DAT-FlpO (*Slc6a3^(em1(flpo)Hbat)*; RRID:IMSR_JAX:035436) mice were generously provided by Dr. Helen Bateup (UC Berkeley). Mice were group-housed and maintained on a 12 h light-dark cycle (i.e., light cycle; 7am-7pm) with food and water available *ad libitum* unless noted. Both male and female mice were included and all experiments were conducted during the light phase of the cycle. Housing and procedure rooms were maintained at a temperature of ~21 C and humidity of ~50%. Animals were on average 15-weeks old at the time of surgery and 28-weeks old by the end of the experiments. When food-restricted for behavioral assays, animals were offered food for 3 hr per day, unless noted. All protocols were approved by the University of California San Diego Institutional Animal Care and Use Committee.

## Stereotactic Surgery

For intracranial injections, mice ( > 6 weeks) were deeply anesthetized with isoflurane, placed into a stereotaxic apparatus (Kopf), and 150-250 nL of AAV5-EF1α-DIO-hChR2(H134R)-mCherry (5.25 ×10^12 GC/ml, RRID:Addgene_20297), AAV5-EF1α-DIO-mCherry (7.3 ×10^12 GC/ml, UNC gene therapy center), AAVDJ-EF1α -fDIO-hChR2-EYFP (2.19 ×10^13 GC/ml, plasmid RRID:Addgene_55639, Vigene production) or AAV5-EF1α-DIO-EYFP (6.5 ×10^12 GC/ml, UNC gene therapy center) were infused unilaterally into the left VP (ML = −1.45, AP = +0.55, DV = −5.3; mm relative to Bregma) using custom-made glass pipettes ( ~ 25um aperture diameter) and a Nanoject III (Harvard Apparatus) at a rate of 10 ul/s with 1-s pulse and 5-s inter-pulse interval. Pipette was held in place for 5 min after injection. 400-500 nL AAV5-hSyn-Flex-GCaMP6f (5.4 ×10^12 GC/ml, U Penn vector core), AAVDJ-EF1a-fDIO-GCaMP7f (5 ×10^11 GC/ml, generously provided by Dr. B.K. Lim, UC San Diego) or AAV5-CAG-dLight1.1 (8.1 ×10^12 GC/ml, RRID:Addgene_111067) were similarly infused unilaterally into the VP, VTA (LM = −0.4, AP = −3.4, DV = −4.4), NAc dmsh (LM = −0.8, AP = +1.54, DV = −4.3), or NAc vmsh (LM = −0.9, AP = +1.5, DV = −4.8). Following viral infusion mice were implanted with a 1.25mm-diameter ceramic ferrule, 6-mm long, 200-µm core, 0.37 NA optic fiber (Hangzhou Newdoon Technology) at one of the following coordinates (in mm relative to Bregma): VP (LM = −1.4, AP = 0.55, DV = −4.7) or VTA (LM = −0.4, AP = −3.4, DV = −4.0) for optogenetic stimulation; and/or with a 1.25mm-diameter metal ferrule, 6-mm long, 400-um core, 0.48 NA optic fiber (Doric Lenses, Canada) at one of the following coordinates: VTA (LM = −0.5, AP = −3.4, DV = −4.2), NAc dmsh (LM = −0.8, AP = +1.5, DV = −4.1), NAc vmsh (LM = −0.85, AP = +1.5, DV = −4.7), or VP (LM = −1.3/−1.4, AP = +0.5, DV = −4.8/−5.0), for fiber photometry recordings. Fibers were stabilized in place using dental cement (Lang dental) and secured by two skull screws (Plastics One). Animals were treated with Carprofen (5 mg/kg s.c.; Rimadyl) prior to and 24 hr after surgery. Mice were allowed to recover from surgery ≥ 6 weeks before experiments began.

## Histology

Mice were deeply anesthetized with sodium pentobarbital (200 mg/kg; i.p.; VetOne) and transcardially perfused with 10 ml of phosphate buffered saline (PBS) followed by ~50 ml 4% paraformaldehyde (PFA) at a rate of 5-6 ml/min. Brains were extracted, post-fixed in 4% PFA at 4 °C overnight, and transferred to 30% sucrose in PBS for 48-72 hr at 4 °C. Brains were flash frozen in isopentane and stored at −80 °C. 30-µm coronal sections were cut using a cryostat (CM3050S, Leica) and collected in PBS containing 0.01% sodium azide. For immunostaining, brain sections were gently rocked 3 ×5 min in PBS, 3 ×5 min in PBS containing 0.2% triton X-100 (PBS-Tx) then blocked with 4% normal donkey serum (NDS) in PBS-Tx for 1 hr at room temperature (RT). Sections were incubated in one or more primary antibody: rabbit anti-GFP (1:2000; Molecular Probes Cat# A-11122, RRID:AB_221569), chicken anti-GFP (1:2000; Thermo Fisher Scientific Cat# A10262,

RRID:AB_2534023), rabbit anti-TH (1:2000; Millipore Cat# AB152, RRID:AB_390204), sheep anti-TH (1:2000; Pel-Freez Biologicals Cat# P60101-0, RRID:AB_461070), rat anti-substance P (1:400; Millipore Cat# MAB356, RRID:AB_94639), or rabbit anti-DsRed (1:2000; Takara Bio Cat# 632496, RRID:AB_10013483) in block at 4 °C overnight. Sections were rinsed 3 ×10 min with PBS-Tx and incubated in appropriate donkey secondary antibodies (Jackson ImmunoResearch Labs) conjugated to Alexa488, Alexa594 or Alexa647 fluorescent dyes ( ~ 5 µg/ml) for 2 hr at RT: Donkey anti-rabbit alexa488 (711-545-152, RRID:AB_2313584), Donkey anti-chicken alexa488 (703-546-155, RRID:AB_2313584), Donkey anti-rabbit alexa594 (711-585-152, RRID:AB_2340621), Donkey anti-rabbit alexa647 (711-605-152, RRID:AB_2492288),Donkey anti-sheep alexa647 (713-605-147, RRID:AB_2340751), Donkey anti-rat alexa647 (712-605-153, RRID:AB_2340694). Sections were washed 3 ×10 min with PBS, mounted on slides, and coverslipped with Fluoromount-G mounting medium (Southern Biotech) containing DAPI (0.5 µg/ml; Roche).

Histochemical characterization was performed on images acquired using a Zeiss AxioObserver Z1 widefield epifluorescence microscope (10×0.45 NA, 20×0.75 NA, or 63×1.4 NA objective) and Zen blue software (ZEN Digital Imaging for Light Microscopy, RRID:SCR_013672). High-magnification display images were acquired with a Zeiss ApoTome 2.0. VP boundaries were defined using Substance P staining. Spread of ChR2 and GCaMP expression, defined by areas containing cell bodies expressing the fluorescent reporter, and optic fiber placements were mapped onto corresponding coronal sections in the Mouse Brain Atlas (Paxinos & Franklin, second edition, 2001 version) using Adobe Illustrator (v.28.3). Mice were excluded when misplacement of one of the optic fibers (on top of NAc, VP, or VTA) was detected (number of animals excluded: 6 VGAT-Cre mice, 9 VGLUT2-Cre mice, 3 VGLUT2-Cre; VGAT-FlpO mice).

## Behavior

**High-fat high-sugar (HFHS) and novel object interactions.** High-fat high-sugar (HFHS) food pellet and novel object interactions were assessed in the context of a previously employed HFHS conditioned place preference (CPP) assay[33] to validate the rewarding properties of HFHS. Briefly, following free exploration of an empty two-sided apparatus during a 20-min pre-test session, mice were confined in one or the other side with either a HFHS pellet or an inert object during 20-min conditioning sessions, followed by a 20-min test session during which mice were able to freely explore the empty apparatus one last time. Time spent in each compartment was recorded using video tracking software (ANY-maze, RRID:SCR_014289). A ~ 2 g-pellet of HFHS (rodent diet with 45 kcal% fat, D12451; Research Diets) was placed in a corner of the side defined as 'paired', and a marble was placed in a corner of the side defined as 'unpaired'. Mice were confined alternatively to both paired and unpaired sides during 2 20-min sessions over 4 days (1 session / day). The assignment of paired and unpaired sides was counter-balanced between animals. Mice were habituated to HFHS food pellets in their home cage, but not to marbles, prior to the beginning of the assay. Mice were food-restricted to 90% of their basal body weight during the HFHS CPP protocol to incentivize HFHS value. Mice were tethered with fiber photometry (FP) optic fiber (400um-core, NA 0.48, DORIC Lenses, Canada) through a pigtailed rotary joint (DORIC) to allow animals free-range of movements during pre-test, all conditioning sessions, and test. Paired and unpaired sessions were counter-balanced between animals. Interactions with HFHS pellet and marble during conditioning sessions, defined as sniffing, contact, chewing, or eating, were manually recorded and time-locked with calcium events. Time in each compartment during the test session was measured as in the pre-test and time spent in the paired compartment was compared between the pre-test and test (delta time (test − pre-test) = 190.8 +/− 60.9 s).

**Strawberry milk licking.** A few drops of strawberry milk (6.25 g of strawberry Nesquik in 100 ml of whole milk) were placed in the home cage the day prior to testing. On the first day, food-restricted mice were placed for 30 min in an operant chamber (Med Associates; controlled by MED-PC IV software, RRID:SCR_012156) with access to a sipper dispensing strawberry milk. On the second day, mice were then tethered with FP patch cord and placed in the same chamber for 30 min with a strawberry milk sipper available. Licks were recorded as previously described[97] and time-locked to calcium events, either for all licks or only licks with the specified minimal and maximal inter-lick interval (ILI; 1 to 2 s, 2 to 5 s, 5 to 10 s, 10 to 20 s, or 20+s). Lack of data for one or more ILI bins resulted in the exclusion of 1 VGAT-Cre and 2 VGLUT2-Cre mice from ILI analysis.

**Chocolate vs. quinine-coated pellet consumption.** Food-restricted and FP patch cord-tethered mice previously habituated to both regular 20-mg precision dustless pellets (F0071; Bio-Serv) and chocolate-flavored pellets (F05301; Bio-Serv), were presented alternately with chocolate-flavored pellets and regular pellets coated with a 30 mM quinine solution in a plastic disposable cage (Innovive) during a single 15-min session. Two to three pellets of one type were presented at a time. Once all were consumed or once the animal stopped approaching them for longer than 2 min, the other type of pellet was presented. Each type of pellet was presented alternatively until the end of the 15-min session. Events were registered to calcium activity and scored as different types: grab-eat (consumption), grab-reject (pellet grabbed and displaced by the animal), and no-grab approach (contact, sniffing without displacing pellet). Events were time-locked to calcium activity.

**Conditioned footshock.** FP patch cord-tethered mice were placed in an operant chamber for 5-min habituation, followed by delivery of ten 500-ms electric shock of 0.6 mA at VI 60 s (45-75 s). Each shock was directly preceded by a 5-s auditory cue at 90 dB and 3 KHz. Mice underwent 2 conditioning sessions on two successive days, followed by an extinction session during which mice were placed in the same chamber and ten 5-s cues were delivered but no longer followed by shock. Cues and shock were time-locked to calcium events.

**Real-time place procedure.** On a baseline (pre-test) day, mice were tethered to a 62.5-μm laser patch cable (custom-made) through an optical commutator (DORIC) and placed on the border between two adjoining (20 × 20 cm) homogenous gray compartments. Time spent in each compartment was recorded using video tracking software (Any-maze). No stimulation was delivered on the pre-test day. On the subsequent day, one side was designated active and entry to the active side triggered a TTL-controlled laser (473-nm DPSS laser, Shanghai laser) to deliver 10 mW (80 mW/mm² at 200-μm fiber tip) pulses at 40 Hz with a 10-ms pulse width controlled by ANY-maze interface for as long as the animal remained in the active side. Sessions lasted for 20 min and the amount of time spent in each compartment, distance traveled, speed, and number of crossings were recorded. A 7-day protocol was used consisting of a pre-test day, 2 days of test (stimulation on side A), and 2 days of 'switch' (stimulation on side B), followed by 1 day of test coupled with fiber photometry recording (side A), and 1 day of 'switch' coupled with fiber photometry recording (side B). On fiber photometry recording days (day 6 & 7), mice were tethered to both a laser patch cord and a FP patch cord connected to independent rotary joints or optical commutators.

**2-nosepoke intracranial self-stimulation (ICSS).** Training. Food-restricted mice were tethered with a laser patch cord and placed in operant chambers. The chamber contained two photobeam-equipped ports which were each baited at the start of each session with a 20-mg precision dustless pellet (Bio-Serv, F0071). During 45-min sessions, nosepokes on the active head-port led to: a 1-s tone (2 kHz), LED cue

lights over the head-ports blinked off for 1 s, and the delivery of a 1-s 40-Hz, 10-ms pulse width, 473-nm laser stimulation controlled by Arduino (Arduino UNO, RRID:SCR_017284). Nosepokes that occurred during the 1-s timeout period of laser stimulation were recorded but were without effect. Inactive nosepokes led to identical tone and cue light effects but did not trigger the laser. Mice were trained over 3 consecutive days to associate a nosepoke in the active head-port with stimulation.

Testing. Following training, mice were fed ad libitum. For testing, mice were connected to both laser patch cord and FP patch cord and placed back in the same operant chamber to record fluorescence in VTA or NAc while animals self-stimulated VP or VP terminals in VTA. Across sessions, increasing durations of timeout (TO) periods were tested (1, 2, 5, 10, and 20 s). TO and availability of the next stimulation were signaled by the LED cue lights over the head-ports turning off for the duration of the TO period. For VP Glut neuron or terminal stimulation, only one day of recording was performed with a 20-s timeout period.

Passive playback (PPB). The temporal pattern of stimulation delivered during each individual subject"s ICSS assay (20-s timeout) was extracted, and Med-PC code was programmed to replay this pattern of laser stimulation during a PPB session. For PPB, mice were placed in the same chamber, nosepokes in either head-port still triggered a 1-s tone with cue lights blinking off for 20 s, but nosepoke in the active head-port no longer triggered laser. Instead, mice received stimulation via passive playback. Stimulation was not signaled by any cues in this protocol. An additional PPB session was performed the following day (not shown), with head-ports no longer available and PPB stimulation signaled by the simultaneous delivery of a 1-s tone. Calcium and DA signals were recorded as in previous conditions.

### Passive/non-behavior contingent stimulations

Animals were placed in a plastic disposable cage (Innovive, CA) and tethered to both laser and FP patch cables for 7 to 14-min sessions, testing a maximum of 4 parameters per session, with 10 trials each. Stimulation parameters tested were: frequency (5, 10, 20, 40 Hz), duration (0.5, 1, 2, 5 s), and inter-stimulation interval (ISI; 2, 5, 10, 20 s). When testing different frequencies and durations, ISI was fixed to 20 s. When testing different ISI, stimulation was set to 40 Hz and 1 s duration. Laser stimulation was controlled by ANY-maze interface.

### Calcium and dopamine fiber photometry

GCaMP6f, GCaMP7f or dLight1.1 were excited by amplitude modulated signals from two light-emitting diodes (465- and 405-nm isosbestic control, DORIC) reflected off dichroic mirrors (6-port minicube, DORIC) and coupled into a single FP optic fiber (400um-core, NA 0.48, DORIC). Sensor signals and isosbestic control emissions were returned through the same optic fiber and acquired using a femtowatt photoreceiver (Newport), digitized at 1017 Hz, and recorded by a real-time signal processor (RZ5P, Tucker Davis Technologies, RRID:SCR_006495). Behavioral and event timestamps were digitized in Synapse software (Tucker Davis Technologies) by TTL inputs from Med-PC, Any-Maze, or manual triggers. Power of light used for imaging (<0.2 mW) is ~2 orders of magnitude less than used for laser stimulation (10 mW), and the distance between the brain sites of stimulation and calcium or DA imaging was >3.5 mm.

### Calcium and dopamine signal analyzes

Analysis of the recorded calcium and DA signals was performed using custom-written MATLAB scripts (MATLAB v2020a, RRID:SCR_001622; with packages: eeglab v.14.1.1b, raacampbell/shadedErrorBar v.1.62.0.0, and chronux v.2.12). Signals (465 and 405 nm) were extracted event-by-event, 5 s before to 10 s after each event was analyzed. For each trial, data were detrended by regressing the isosbestic control signal (405 nm) on the sensor signal (465 nm) and then

generating a fitted 405-nm signal using the linear model generated during the regression. ΔF/F was obtained by subtracting the fitted 405-nm channel from the 465-nm signal, and by dividing the resulting by the fitted 405-nm signal, to remove the potential movement, photobleaching, or fiber bending artifacts (ΔF/F = (465 signal − fitted 405 signal) / fitted 405 signal[98];). Normalized ΔF/F for the 15 s peri-event time windows (ΔF/F %; in Supplementary Figs. 4, 6-7 & 9) were calculated as follow for each trial: ΔF/F − (mean (baseline)), where the baseline was defined as the −1 to 0 s pre-event time window, except for Fig. S4G, H, S7G–I (−0.1 to 0 s), and Fig S4I, Fig S6C–E (−2 to −1s). Z-scores were calculated as follows for each trial: (ΔF/F − mean (baseline)) / (standard deviation (baseline)), where the baseline was defined as the −5 to 0 s pre-event time window. Normalized ΔF/F and z-score traces were generated in MATLAB. Normalized ΔF/F are represented as mean and z-score as mean +/- SEM in figures. Then z-scores were extracted per 1-s time window from −2 to 7 s around the event, and averaged per animal. Peak z-score (highest absolute value with reassigned positive or negative sign) during specified 1-s time windows (baseline: −2 to −1s; pre-event: −1 to 0 s; event/stimulation/early stimulation: 0 to 1 s; late stimulation: 4 to 5 s; cue: 0 to 1 s; shock: 5 to 6 s) were then extracted per animal and averaged per group. Max peaks (highest value) and min peaks (lowest value) during stimulation (0 to 1 s) were also extracted for VTA GABA responses to VP GABA stimulation (Fig. 4F). Peak (as well as max and min) z-score histograms were generated in GraphPad Prism (GraphPad Prism v6.01, RRID:SCR_002798).

### Ex vivo electrophysiological recordings

Adult mice (7–12 weeks) were deeply anesthetized with pentobarbital (200 mg/kg i.p.; Virbac) and perfused intracardially with 10 ml ice-cold sucrose-based ACSF containing the following (in mM): 75 sucrose, 87 NaCl, 2.5 KCl, 7 MgCl2, 0.5 CaCl2, 1.25 NaH2PO4, 25 NaHCO3, and continuously bubbled with carbogen (95% O2 − 5% CO2). Brains were extracted, and 200 µm coronal slices were cut in sucrose-ACSF using a Vibratome (vt1200S, Leica). Slices were then transferred to a perfusion chamber containing ACSF at 31 °C (in mM) as follows: 126 NaCl, 2.5 KCl, 1.2 MgCl2, 2.4 CaCl2, 1.4 NaH2PO4, 25 NaHCO3, 11 glucose, continuously bubbled with carbogen. After at least 45 min, slices were transferred to a recording chamber continuously perfused with ACSF (2–3 ml/min) and maintained at 29–31 °C using an inline heater. Patch pipettes (3.5–5.5 MΩ) were pulled from borosilicate glass (King Precision Glass) and filled with internal recording solution containing the following (in mM): 120 CsCH3SO3, 20 HEPES, 0.4 EGTA, 2.8 NaCl, 5 TEA, 2.5 Mg-ATP, 0.25 Na-GTP, at pH 7.25 and 285 mOsm. Before recordings, 0.1% Lucifer yellow (Sigma) was added to the internal recording solution.

mCherry-labeled terminals were visualized by epifluorescence, and visually guided patch recordings were made from VTA cells using infrared-differential interference contrast illumination (Axiocam MRm, Examiner.A1, Zeiss). ChR2 was activated by flashing blue light using a light-emitting diode (UHP-LED460, Prizmatix) under computer control. EPSCs were recorded at −70 mV of holding in whole-cell voltage clamp (Multiclamp 700B amplifier), IPSCs at 0 mV, filtered at 2 kHz, digitized at 10 kHz (Digidata 1550) and collected on-line using Clampex 10.6 software (Molecular Devices). Series resistance and capacitance were electronically compensated before recordings. Estimated liquid-junction potential was 12 mV and left uncorrected. Series resistance was monitored and cells that showed >25% change in current baseline during recordings were discarded. For whole-cell voltage-clamp recordings, single-pulse (5 ms) photostimuli were applied every 55 s and 10 photo-evoked currents were averaged per neuron per condition; current sizes were calculated by using peak amplitude from baseline. For cell-attached studies on firing rate, photostimuli trains (5 s) were delivered every 45 s and 3 responses averaged per neuron. Action potential frequency was averaged over the 1 or 5 s before,

during, and after the stimulation train. DMSO containing DNQX (Sigma) was diluted 1000-fold in ACSF and bath applied at final concentration of 10 µM. PSCs amplitude, frequency of firing rate, and representative traces were analyzed and extracted using Clampfit software v10.4.2 (Molecular Devices).

To assess whether the recorded VTA neurons were dopaminergic, we performed post hoc immunostaining for tyrosine hydroxylase (TH) on 200-µm slices used for whole cell patch clamp electrophysiology. Slices were postfixed with 4% PFA overnight and then washed three times (5 min) in PBS, three times (5 min) in PBS containing 0.2% Triton X-100 (PBS-Tx), and blocked in PBS-Tx containing 4% normal donkey serum (NDS) for 1 hr at room temperature. Free-floating slices were then incubated in primary antibody against TH (rabbit anti-TH, 1:1000, Millipore AB152) 48 hrs at 4 °C overnight. Sections were washed again three times (10 min) in blocking solution (PBS-Tx) and incubated with secondary antibody (donkey anti-rabbit conjugated to Alexa-647 fluorescent dye, 5 µg/ml, Jackson ImmunoResearch Laboratories) for 24 h at 4 °C. Slices were rinsed three times (10 min) with PBS and mounted onto slides with Fluoromount-G mounting medium (Southern Biotech) ± DAPI (Roche, 0.5 µg/ml). Images were acquired using a Zeiss Axio Observer equipped with ApoTome, and visualized using Zeiss Zen software (Zeiss). Images acquired during recording and post-immunostaining were compared to validate the location of the recorded cell. Lucifer yellow cells were identified by their yellow fluorescence and were considered TH+ if fully overlapping with alexa-647 signal.

### Statistical analysis

Behavioral, electrophysiological, and FP z-score datasets were analyzed by two- or one-tailed t-tests, repeated measure (RM) one-way ANOVAs, two-way ANOVAs, or three-way ANOVAs as appropriate (full statistical reporting for main text Figures available in Table S2, and in legends of Supplementary Figs.) using GraphPad Prism (GraphPad Prism v6.01, and GraphPad Prism v9.2 for three-way ANOVAs, RRID:SCR_002798). Linear regression was calculated between event peak z-scores and ILIs for the strawberry milk-licking assay. For FP z-score datasets, statistical comparisons were only performed within a given preparation (i.e., mouse line, opsin, and sensor) to avoid potential confounds when comparing across these parameters, e.g., due to variable numbers of cells or expression levels. Both male and female mice were used, but the sample size was not powered to examine sex differences. Sex was, therefore, not included as a factor in statistical analyses. However, individual data are disaggregated by sex in Table S2 and the Source data files.

### Reporting summary

Further information on research design is available in the Nature Portfolio Reporting Summary linked to this article.

## Data availability

Source data are provided with this paper.

## Code availability

Custom-written MATLAB basic codes for FP analysis are available on github at https://github.com/lauren-faget/VPtoVTA_FP.

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

## Acknowledgements
We thank Dr. B.K. Lim (UCSD) for providing the fDIO-GCAMP7f virus. This work was supported by grants from the National Institutes of Health (NIH) (R21MH118748, T.S.H. and L.F.; R01DA036612, T.S.H.), and the Veterans Affairs (VA) San Diego Health Systems (I01BX005782, T.S.H.).

## Author contributions
Conceptualization: L.F. and T.S.H.; Methodology: L.F. and T.S.H.; Investigation: L.F., L.O., W.C.L., V.Z., C.S., and A.F.; Formal analysis: L.F., L.O., W.C.L., V.Z., D.R. and T.S.H.; Writing - original draft: L.F. and T.S.H.; Writing - review and editing: L.F., N.H., T.S.H.; Funding acquisition: L.F. and T.S.H.

## Competing interests
The authors declare no competing interests.
