## [Peer Review File · Nature Communications]

Ventral pallidum GABA and glutamate neurons drive approach and avoidance through distinct modulation of VTA cell typesREVIEWER COMMENTS

Reviewer #1 (Remarks to the Author):

This study by Faget and colleagues is an interesting follow-up on a previous article by this group that showed that ventral pallidum glutamate (VP Glut) and VP GABA neurons oppositely regulate reinforcement. Here, the authors show that the optogenetic stimulation of these cells produces differential calcium activity in ventral tegmental area dopamine (VTA DA), glutamate (VTA Glut) and VTA GABA neurons, and that these signals are attenuated by reward expectancy in experienced animals. The authors also show that VP GABA and Glut neurons may oppositely regulate motivated behaviors through their differential recruitment of VTA dopamine efferents to the dorsomedial versus ventromedial nucleus accumbens shell. Overall, this is an interesting paper with exciting findings. I do, however, have a few comments and concerns regarding the used methodology and interpretation of results.

Major:

A main limitation of the currently used approach (cell body stim of VP neurons undefined by projection target) is that calcium activity recorded at the level of the VTA during VP stim is not necessarily regulated by VP-VTA projections, and may be (co-)regulated by other intermediary structures (e.g. local VP collaterals, VP projections to the lateral habenula, lateral hypothalamus, rostromedial tegmentum, etc). Unfortunately, the temporal resolution of photometry is not fast enough to distinguish between mono- and polysynaptic connections in the circuit. At the very least this should be addressed in more detail in the discussion, and the proposed model (Fig. 7) is likely substantially more complex than depicted. Except for the final experiments, where VP-VTA axons were directly stimulated, it would be best to omit arrows from experimental diagrams in figures (or add additional arrows). The model in Fig. 7 also appears to suggest that VP GABA neurons preferentially innervate VTA GABA neurons, but other work suggests that this is not the case (e.g. PMID: 32541962). To strengthen the proposed model, the authors could consider using complementary methods (e.g. whole cell recordings, or synaptotag quantification of axonal inputs from VP GABA to different VTA cell types) to show a preferential (strength of) monosynaptic connection between VP GABA and VTA GABA neurons.

The finding that goal-directed actions attenuate the responses of VTA cell types to VP stimulation in experienced animals is very interesting and appears to indicate reward expectation as the authors suggest. To support this conclusion, it would be helpful to show in a group of naïve mice that the signal changes as animals become familiar with the task (e.g. RTPP) and come to expect the opto stim.

Minor:

For the 'punishment' experiment where HFHS pellets were coated with quinine it would be helpful to show the reported decrease in consumption over time. Also, were there any changes in the recorded calcium signals as consumption decreased?

It would be useful to see whether any relationships (correlations) exist between the behavioral responses of individual animals and recorded magnitude of their calcium activity.

For the experiment with the lickometers (Fig. 1E, F) it is unclear how interval durations were binned. Were all events with intervals y larger than x and smaller than z (i.e. $x < y < z$) combined?

It would be helpful to quantify the rebound 'excitation' seen in some of the recordings and discuss the potential functional significance of this signal in the discussion.

It would be useful to cite Zhu et al. who first described the reinforcing effects of optogenetic VP GABA stim (PMID: 28683305), and Soden et al. who showed that VP GABA projections to the VTA are more prominent than VP Glut projections, and that VP GABA projections to the VTA do not bias towards VTA GABA neurons (as other forebrain GABA projections do; PMID: 32541962).

Reviewer #2 (Remarks to the Author):

The present manuscript from Faget et al. expands on previous work from the group on ventral pallidum's circuitry and how it participates to reward and aversion processing. Using fiber photometry and optogenetics, this study aims at dissecting how VP GABA and glutamate projections to the VTA differentially participate to approach and avoidance behavior.

The results presented in this set of experiments are very interesting and constitute an important contribution to the field. The manuscript is well written, and the different experiments and related results are clearly described. Below are some comments and recommendations:

- 1) The way the authors dissected the activity of the different VTA cell types in response to VP GABAergic or glutamatergic stimulation is very informative. Overall, they suggest that for reward, VP GABA neurons inhibit VTA GABA neurons, ultimately allowing dopamine release in both dorsal and ventral NAc mshell.

On the other hand for aversion, VP glutamate neurons would activate VTA GABA neurons resulting in inhibition of dopamine release in dorsal NAc mShell. It thus seems that VTA GABA neurons constitute a key node, and their activation or inhibition is crucial for driving reward and aversion.

Here the activity pattern of VP and VTA neuron subtypes is mostly derived from observations of neuronal activity pattern associated with optogenetic stimulation during several tasks, for example optogenetic stimulation of VP neurons during RTPP. However, VP glutamatergic and GABAergic neurons project to multiple brain regions. To confirm the key role of VTA GABAergic neurons in reward and aversion it would be important that the authors manipulate the VP GABA/glutamate → VTA GABA → VTA dopamine circuit. One option would be to measure behavior while altering this circuit by inhibiting VTA GABAergic neurons (using DREADDs for example) during VP glutamatergic neuron stimulation. This could contribute to confirm the key role of VTA GABA neurons in reward vs. aversion.

2) The authors collected a wide range of very interesting data spanning from event-induced to optogenetic stimulation-induced neuronal activity. When reading the manuscript, this variety of activation pattern is sometimes difficult to apprehend. For instance, in the fear conditioning experiment, VP GABA neurons are active during both the US and CS while VP glutamatergic neurons are only active during the US. In the manuscript, it is unclear how this relates to the idea that "...VP GABA neurons respond to reward and their activity can drive reward seeking, while VP Glut neurons respond to aversive stimuli and drive avoidance behavior" (page 15-16). Could the authors discuss and compare a little bit more the findings from experiments assessing firing during "naturalistic" behaviors to those obtained with optogenetic stimulation?

3) Figure 7A summarizes well the author's idea of the mechanisms at play between the two VP neuronal subpopulations. However, this figure is based on neuronal response at the end of a 5 sec passive stimulation. At several steps of their work, the authors show that active, voluntary, stimulation is associated with smaller or slightly different responses in the VTA compared to passive stimulation. For instance, VP GABA ICSS induces a brief activation followed by a small inhibition of VTA GABA neurons (later followed by a rebound of activity). It would be interesting if the authors could discuss that aspect when referring to Fig 7A.

4) Figure 7B is very helpful to summarize the mechanisms described by the authors. It clearly shows the differences between the GABA and glutamate circuits. However, since approach of both rewarding and aversive stimuli elicits both GABA and glutamate VP neuron activation, an additional figure (or combining information from Fig 7A and 7B) showing activation patterns of both circuits, in parallel, during reward and aversion processing could be very informative. For example: what happens in both circuits for reward vs. what happens in both circuits for aversion? This could give a broader overview of the neuronal mechanisms studied here.

Minor comments

A) The authors mention that both males and females were used. In the different figures, could the authors differentiate both sexes (using different symbols when showing individual data points for example)?

B) In the Results section (last paragraph of page 9 of the pdf file), the authors wrote: "... again suggesting that VP GABA stim when evoked by the subjects' action inhibited the resulting VTA DA neuron response". The use of the word "inhibited" seems a little bit strong and could be replaced by a word such as "reduced" for example.

C) In the Methods section, regarding "Chocolate vs. quinine-coated pellet consumption", could the authors add a few more details on how pellets were alternatively presented? For example, did the authors presented 10 chocolate pellets and then 10 quinine-coated pellets? Or did the authors alternate between both flavor?

D) Throughout the manuscript the authors use "stim" to refer to optogenetic stimulation. As this does not impact the manuscript word count, could the authors use "stimulation" in the text to facilitate reading?

Reviewer #3 (Remarks to the Author):

In this manuscript, Faget et al. examine the activity patterns of ventral pallidal (VP) GABA and glutamate neurons during approach and aversion, and the impact of activity in these populations on downstream activity in distinct VTA cell types, and dopamine release in subregion of the nucleus accumbens. They find that population calcium activity in both GABA and glutamate neurons in VP increases during approach to food reward, as well as a novel object (though this response decreases over time). Interestingly only GABA neurons were responsive to a cue paired with shock, whereas both cell populations responded to the shock itself. The impact of VP GABA neurons stimulation varies based on VTA cell type, as well as the animals' control over the stimulation. The impact of VP glutamate neurons stimulation is more consistent across VTA cell types and is less sensitive to the animal's agency. While both GABAergic and glutamatergic VP to VTA neurons can evoke increased dopamine neurons activity on a population level, glutamate neuron activation has opposing effects on dopamine release in the ventral versus dorsal subregion of the medial shell, which have been differentially implicated in reward and aversion. The data presented here are comprehensive and will be very informative to researchers trying to understand pallidal-midbrain signaling and control of motivated behavior. The statistical analysis is

largely appropriate and rigorous, and the manuscript is clearly written and scholarly. I have largely minor concerns and comments, listed below.

1. In Figure 1E-F, the authors compared the magnitude of calcium activity depending on the minimal inter-lick intervals (ILIs). Larger minimal ILIs were associated with greater peak z-scored fluorescence, which they interpreted as VP GABA activity being preferentially responsive to the initial approach to an appetitive stimulus, but I'm concerned that this may be due in part to differences in the activity during the baseline period that is used for z-scoring (0-5s pre-event). It seems like licking during this baseline period when ILIs is defined as 1-2 could be artificially reducing the maximum z-score, based on a greater STD during this period. Does the pattern reported here hold true even if data is plotted as dF/F , rather than z-scored to this baseline?

2. The similar responses to both chocolate and quinine pellets in Figure 2 is somewhat surprising. Did the quinine responses change across the session as animals transitioned from consuming the pellets to only sniffing, but not consuming them?

3. It would be informative to know whether the cue responses in VP GABA neurons during footshock conditioning emerged following learning, or were consistent across conditioning. Was the cue response present on trial 1 (or trials 1-5)? If the cue response was present early on did it diminish at all across conditioning trials, potentially suggesting more of a novelty signal? Knowing the time course would help aid interpretation of this signal.

Reviewer #4 (Remarks to the Author):

This work by Faget and colleagues comprises a series of comprehensive studies that advance understanding of how different populations of neurochemically-defined subtypes in the ventral pallidum orchestrate appetitive and avoidance behavior. The complementary approaches corroborate that the discrete effects of GABAergic and glutamatergic (VPGABA and VPGlu, respectively) VP neurons on appetitive behavior are driven – at least in part – by unique post-synaptic targets in the VTA, rather than the simple model previously posited that VP-GABA neurons inhibit while VP-Glu neurons excite the lateral habenula and RMTg, respectively. The dissociation of dopamine neurons projecting to the dorsal- vs. ventral-medial accumbens shell is particularly elegant in this regard. The finding that signaling in these pathways and in their ability to recruit post-synaptic targets in the VTA depends on whether actions are volitional is also novel, and is consistent with a growing body of evidence pointing to a role of the ventral pallidum in the calculation of reward prediction error.

While somewhat descriptive in nature, the findings nonetheless significantly advance models of how the VP orchestrates appetitive behavior. The experiments are straightforward in their design and analysis, and the technical limitations of the approaches are thoroughly discussed. My main suggestion for improvement refers to improving the quality of the schematic in the final figure. This serves as a summary of the many findings of the paper, and more attention to the level of detail in the circuit diagrams would help readers grasp the anatomical and functional conclusions of the manuscript.

We thank the Reviewers for their thorough evaluations of our manuscript. We were pleased by the general enthusiasm expressed by Reviewers. We have amended the manuscript to include new data, analysis, figures (**Figures S2, S3 and S5**) and other figure edits and text revisions as detailed in the below point-by-point response.

Reviewer #1:

This study by Faget and colleagues is an interesting follow-up on a previous article by this group that showed that ventral pallidum glutamate (VP Glut) and VP GABA neurons oppositely regulate reinforcement. Here, the authors show that the optogenetic stimulation of these cells produces differential calcium activity in ventral tegmental area dopamine (VTA DA), glutamate (VTA Glut) and VTA GABA neurons, and that these signals are attenuated by reward expectancy in experienced animals. The authors also show that VP GABA and Glut neurons may oppositely regulate motivated behaviors through their differential recruitment of VTA dopamine efferents to the dorsomedial versus ventromedial nucleus accumbens shell. Overall, this is an interesting paper with exciting findings. I do, however, have a few comments and concerns regarding the used methodology and interpretation of results.

Major:

A main limitation of the currently used approach (cell body stim of VP neurons undefined by projection target) is that calcium activity recorded at the level of the VTA during VP stim is not necessarily regulated by VP-VTA projections, and may be (co-)regulated by other intermediary structures (e.g. local VP collaterals, VP projections to the lateral habenula, lateral hypothalamus, rostromedial tegmentum, etc). Unfortunately, the temporal resolution of photometry is not fast enough to distinguish between mono- and polysynaptic connections in the circuit. At the very least this should be addressed in more detail in the discussion, and the proposed model (Fig. 7) is likely substantially more complex than depicted. Except for the final experiments, where VP-VTA axons were directly stimulated, it would be best to omit arrows from experimental diagrams in figures (or add additional arrows). The model in Fig. 7 also appears to suggest that VP GABA neurons preferentially innervate VTA GABA neurons, but other work suggests that this is not the case (e.g. PMID: 32541962).

We fully agree with the limitation raised by the Reviewer. We have modified our schematics in relevant figures in the Results section as suggested. We have also altered summary Figure 7, and we increased our discussion of this limitation (lines 388-398). We do wish to emphasize that Figure 7 is a proposed circuit model, called out from the Discussion section, and we expect that follow-on experiments will amend and embellish the proposed model. This too is now clearly stated in the discussion (line 419).

To strengthen the proposed model, the authors could consider using complementary methods (e.g. whole cell recordings, or synaptotag quantification of axonal inputs from VP GABA to different VTA cell types) to show a preferential (strength of) monosynaptic connection between VP GABA and VTA GABA neurons.

Prior to embarking on the studies included in the initial submission, we performed *ex vivo* electrophysiological recordings in VTA neurons with optogenetic stimulation of VP terminals. VTA neurons were identified as DA or non-DA by post-recording TH immunostaining. Results from this experiment have now been added to the revision (**Supplemental Figure 3**). Using whole cell voltage-clamp or cell-attached recordings of spontaneous firing, these data indicate that VP GABA and VP glutamate projections evoke inhibitory or excitatory responses, respectively, in both DA and non-DA VTA neurons. If anything, the data may suggest that VP GABA neurons synapse preferentially on to non-DA neurons, while VP glutamate neurons preferentially synapse on to DA neurons. However, the results were variable (not black and white), and it was not possible to rule out spread of opsin to neighboring structures in this preparation. At the time of their collection, these data left us with more questions than answers, particularly, how to understand why VP glutamate neurons, which drive strong avoidance behavior, activate VTA DA neurons. We now understand that this could be explained by VP glutamate recruitment of VTA DA subpopulations associated with aversion. While one could take this approach further, we ultimately decided that this *ex vivo* approach wouldn't allow us to understand how activity of VP cell types relate to VTA cell type activity in the context of behavior. That is, regardless of whether the effects of VP activation are monosynaptic or polysynaptic in nature, we were most interested in how cell-type-specific VP activity regulates

activity of VTA cell types in the context of ongoing network activity during behavior. Thus, the *in vivo* calcium sensing experiments presented in this manuscript were designed to collect this data at the integrated circuit level *in vivo*.

The finding that goal-directed actions attenuate the responses of VTA cell types to VP stimulation in experienced animals is very interesting and appears to indicate reward expectation as the authors suggest. To support this conclusion, it would be helpful to show in a group of naïve mice that the signal changes as animals become familiar with the task (e.g. RTPP) and come to expect the opto stim.

Thanks for this suggestion. We agree that such data would be a valuable addition to our manuscript. We therefore attempted to pilot the experiment proposed by the Reviewer, but have identified major challenges, described below.

For this pilot we used a slightly different preparation than used in the prior experiments. The original experiments were designed with stimulation training periods during which no recordings were made. This allowed mice to learn the task contingencies without the constraint of dual cables, but the downside is that we didn't capture photometry responses as mice learned the context-stimulation or operant-stimulation associations. For this pilot, rather than relying on separate fibers for optogenetic stimulation and recording by fiber photometry, we used a single fiber approach because we thought it would give us a better chance of monitoring neural activity while mice were learning new task contingencies. We expressed Chrimson in VP GABA neurons, GCaMP in VTA DA neurons, and implanted only one optic fiber over VTA to stimulate the Chrimson terminals with red laser while recording from GCaMP signals with blue LED. We then tested them on the RTPP and ICSS tasks. The mice behaved as expected, they showed preference for the laser-paired chamber and optogenetic stimulation evoked increased VTA DA neuron calcium signals. We also observed smaller responses when rewards were self-delivered, for example by ICSS compared to passive playback (**Figure A**). And we also tested them in a slightly different version of this assay where they received non-contingent stimulations interspersed during the fixed interval period of an ICSS session, and again we saw the same pattern (**Figure B**). Thus, these results add further support to our main conclusions and suggest that pathway-specific optogenetic stimulation of VP terminals in VTA can explain our main findings, at least with respect to DA neurons, in a manner very similar to the dLight experiments presented in **Figure 6**.

Figure A. VTA DA neuron activity in response to ICSS, passive playback (PPB) or **(B)** variable interval passive stim (VI) of VP GABA neuron terminals in the VTA. ICSS and PPB were performed during 2 consecutive days, while animals could self-stimulate and receive passive stimulation during the session of day 6 and 7 (n=2). The sample size is small because of low availability of VGAT-Flp x DAT-Cre mice and failure to breed efficiently, plus several additional injected mice failed to show an evoked response suggesting failure of Chrimson expression (histology in progress).

However, from these pilots it became apparent that it would be difficult to definitively answer the Reviewer's main question using this approach (**Figure C**). A major challenge is that the learning (expectation) may happen after a variable number of optogenetic stimulus pairings/exposures from mouse-to-mouse, but then very quickly (steeply). For the RTPP experiment mice spend variable durations of time in each chamber, often making only brief entries into each compartment, especially during the earlier phases of the task when exploratory behavior (and presumably learning) is

highest. However, we can't readily quantify photometry signals when mice were rapidly moving between chambers, staying <5s in the prior side, due to decay kinetics and carry-over effects of the GCaMP signals. Moreover, there is a fair amount of trial-to-trial variability in the photometric responses and thus if learning is happening steeply over a few trials, the diminution of signals with learning could be lost in the 'noise'. We can think of ways to explore this issue further, but it would require validating a new task and likely troubleshooting task variations or different analysis methods. Moreover, it is also possible that the reduction in signal has less to do with expectation and more to do with a process correlated with expectation, such as the goal-directed action itself or moment-to-moment changes in motivation or decision-making that drive instantaneous pursuit behaviors. We ultimately think this is an important question and one that would be better explored in a more comprehensive follow-on investigation, rather than in the present study that is primarily focused on comparing VP GABA and glutamate neurons.

Figure C. Evolution of VTA DA neuron responses during behavior-contingent activation of VP GABA terminals in VTA in the RTPP assay. Individual data per trial, session and animal ($n=2$) represented as peak dF/F (%) during the first 1s from paired side entrance in (a) all trials or (b) trials with a minimum of 2s in the non-laser side prior to entry into laser side, and a minimum of 5s in the laser side. Average number of trials per session (a) 33 ± 4 and (b) 9 ± 1 .

Minor:

For the 'punishment' experiment where HFHS pellets were coated with quinine it would be helpful to show the reported decrease in consumption over time. Also, were there any changes in the recorded calcium signals as consumption decreased?

It would be useful to see whether any relationships (correlations) exist between the behavioral responses of individual animals and recorded magnitude of their calcium activity.

We thank the Reviewer for this question. While the real-time observer made general notes while scoring events, they didn't distinguish between various types of consummatory and approach events in real-time. However, we were able to go back to video and distinguish between 'grab-eat', 'grab-reject', and 'no-grab approach' (including sniff/lick) events. These new analyses were normalized to the number of events each subject engaged in and plotted across session time (**Supplemental Figure 2**). The data show that for the quinine-coated pellets the grab-eat events decreased over time while the grab-reject events increased with time. The data also show there is no major change in mean event amplitude as a function of time (though there may be a reduction in variance).

For the experiment with the lickometers (Fig. 1E, F) it is unclear how interval durations were binned. Were all events with intervals y larger than x and smaller than z (i.e. $x < y < z$) combined?

For the strawberry milk drinking assay, all licks separated by a minimum duration x (1, 2, 5, 10, or 20s) were combined in our original analysis. For example, licks with 1s ILI included licks with 2, 5, 10 and 20s ILI too. However, we now see that this creates problems for statistical analysis. Thus we reanalyzed these data, and are now presenting events with ILI larger than x and smaller than y . Therefore we compared licks with ILI between 1 and 2s, 2 and 5s, 5 and 10s, 10 and 20s,

and larger than 20s to each other. The graphs and statistical differences are similar to what was observed previously and support identical conclusions. The Methods have been updated accordingly (lines 599-601).

It would be helpful to quantify the rebound ‘excitation’ seen in some of the recordings and discuss the potential functional significance of this signal in the discussion

Quantification of rebound excitations has now been added as new analysis in new **Supplemental Figure 5**. While we don’t know what they mean, or if they are physiologically relevant, we have added some discussion on these responses to the Discussion (see lines 468-474).

It would be useful to cite Zhu et al. who first described the reinforcing effects of optogenetic VP GABA stim (PMID: 28683305), and Soden et al. who showed that VP GABA projections to the VTA are more prominent than VP Glut projections, and that VP GABA projections to the VTA do not bias towards VTA GABA neurons (as other forebrain GABA projections do; PMID: 32541962).

Both references have been added.

Reviewer #2:

The present manuscript from Faget et al. expands on previous work from the group on ventral pallidum’s circuitry and how it participates to reward and aversion processing. Using fiber photometry and optogenetics, this study aims at dissecting how VP GABA and glutamate projections to the VTA differentially participate to approach and avoidance behavior. The results presented in this set of experiments are very interesting and constitute an important contribution to the field. The manuscript is well written, and the different experiments and related results are clearly described. Below are some comments and recommendations:

1) The way the authors dissected the activity of the different VTA cell types in response to VP GABAergic or glutamatergic stimulation is very informative. Overall, they suggest that for reward, VP GABA neurons inhibit VTA GABA neurons, ultimately allowing dopamine release in both dorsal and ventral NAc mshell. On the other hand for aversion, VP glutamate neurons would activate VTA GABA neurons resulting in inhibition of dopamine release in dorsal NAc mShell. It thus seems that VTA GABA neurons constitute a key node, and their activation or inhibition is crucial for driving reward and aversion.

Here the activity pattern of VP and VTA neuron subtypes is mostly derived from observations of neuronal activity pattern associated with optogenetic stimulation during several tasks, for example optogenetic stimulation of VP neurons during RTPP. However, VP glutamatergic and GABAergic neurons project to multiple brain regions. To confirm the key role of VTA GABAergic neurons in reward and aversion it would be important that the authors manipulate the VP GABA/glutamate -> VTA GABA -> VTA dopamine circuit. One option would be to measure behavior while altering this circuit by inhibiting VTA GABAergic neurons (using DREADDs for example) during VP glutamatergic neuron stimulation. This could contribute to confirm the key role of VTA GABA neurons in reward vs. aversion.

We thank the Reviewer for the thoughtful critique and positive feedback. VTA GABA neurons indeed seem to constitute a key node in how VP (and other) inputs to VTA regulate other VTA cell types and behavior. We very carefully considered the DREADD-based experiment proposed by Reviewer #2 to silence/inhibit VTA GABA neurons in combination with the optogenetic stimulation of VP neuron types. The core hypothesis built into this experiment is that eliminating VTA GABA neurons from the equation may block the reinforcing effects of VP GABA neuron reinforcement and/or block the aversive effects of VP glutamate neuron stimulation. However, we have some major reservations regarding this experiment that we would like to highlight.

- (i) The experiment would require many animals, resources, and time: including 3 VP groups (GABA opsin, Glutamate opsin, combined non-opsin control to assess the effects of DREADD manipulation alone) x 2 treatment groups (DREADD ligand, vehicle) = 6 groups. The experiment may also engender follow-on requirements, such as to provide photometric or other physiological evidence to demonstrate the effectiveness of the DREADD manipulation. The endeavor is thus a large one, and one we would only pursue if the associated caveats are few.
- (ii) However, previous literature showed that manipulating VTA GABA neuron activity can have effects on behavior, independent of VP manipulations, including inducing aversion and modulating/inhibiting reward seeking behaviors

(Wakabayashi et al., 2019, PMID: 29875446; Zhou et al., 2022, PMID: 36260661; Brown et al., 2012, PMID: 23178810, Van Zessen et al., 2012, PMID: 22445345, Eshel et al., 2015, PMID: 26322583). Thus inhibiting VTA GABA neurons with DREADDs will change the basal state of the animal, requiring non-opsin controls be included, and potentially confounding interpretability of the results (see related point iv).

- (iii) Gi DREADDs may or may not effectively inhibit GABA release from VTA GABA neurons in the face of our optogenetic manipulations of VP GABA and glutamate inputs. It is instead quite possible that while DREADD inhibition reduces baseline activity of VTA GABA neurons, optogenetic activation of either VP GABA or Glutamate neurons will still evoke robust decreases or increases from that shifted baseline. In this eventuality the optogenetically-induced behaviors would not be abolished and the data would not allow us to confirm the key role of VTA GABA neurons in reward and aversion.
- (iv) If VTA GABA neuron activity reduces reward seeking, per the literature cited above, then their DREADD-mediated inhibition may increase reward seeking. In that case the reinforcing effects of VP GABA neuron stimulation may be greater because of the change in state caused by inhibition of VTA GABA neuron inhibition. Thus, alternate outcomes and interpretations may ultimately prove rather more subjective than definitive. We think this is a major confound that greatly dampens our enthusiasm for this approach.
- (v) Finally, the direct projection of VP glutamate neurons to NAc vmsh-projecting VTA DA neurons appears to play an important role, promoting aversion due to increase in DA release in NAc vmsh. This effect is predicted to be independent of VP glutamate neuron effects on VTA GABA neurons, thus inhibiting VTA GABA neurons might have little effect on the avoidance response elicited by VP glutamate neuron activity.

Therefore, while the experiment is of interest and has merit, we ultimately think it has the potential to produce effects in either direction (or no effects, per points iv and v) and could be difficult to interpret. It would also require a large initial and likely follow-on investment in time and resources. For these reasons we think the experiment is better suited for a separate report.

2) The authors collected a wide range of very interesting data spanning from event-induced to optogenetic stimulation-induced neuronal activity. When reading the manuscript, this variety of activation pattern is sometimes difficult to apprehend. For instance, in the fear conditioning experiment, VP GABA neurons are active during both the US and CS while VP glutamatergic neurons are only active during the US. In the manuscript, it is unclear how this relates to the idea that "...VP GABA neurons respond to reward and their activity can drive reward seeking, while VP Glut neurons respond to aversive stimuli and drive avoidance behavior" (page 15-16). Could the authors discuss and compare a little bit more the findings from experiments assessing firing during "naturalistic" behaviors to those obtained with optogenetic stimulation?

We agree that it is somewhat difficult to reconcile our findings that VP GABA and glutamate neurons are both activated by natural appetitive and aversive reinforcers with the findings that their evoked activity has opposite effects on approach and avoidance behaviors. A possibility we propose is that VP GABA and glutamate neurons are similarly responsive to rewarding/aversive stimuli but drive different behavioral responses to those stimuli through differential recruitment of VTA cell types. This might reflect the role of VP (and other forebrain inputs to VTA) in controlling approach/avoid behaviors in the face of conflicting urges (dangerous, risky) circumstances, with the circuits in a sort of competition. Though this is our working hypothesis, there are also other explanations. For example, there may be different subpopulations of VP GABA and VP glutamate cells that contribute to the measured and evoked signals. Or data could reflect inherent limitations to the calcium imaging or optogenetic manipulation. We have edited the statement noted by the Reviewer and discuss these possible explanations (lines 377-387).

3) Figure 7A summarizes well the author's idea of the mechanisms at play between the two VP neuronal subpopulations. However, this figure is based on neuronal response at the end of a 5 sec passive stimulation. At several steps of their work, the authors show that active, voluntary, stimulation is associated with smaller or slightly different responses in the VTA compared to passive stimulation. For instance, VP GABA ICSS induces a brief activation followed by a small inhibition of VTA GABA neurons (later followed by a rebound of activity). It would be interesting if the authors could discuss that aspect when referring to Fig 7A.

We thank the Reviewer for this really interesting comment. We have edited **Figure 7A** to reflect the entire responses (AUC over the 5s of stimulation) to both passive and self-administered stimulations in the RTPP assay, and have added

discussion (lines 403–405).

4) Figure 7B is very helpful to summarize the mechanisms described by the authors. It clearly shows the differences between the GABA and glutamate circuits. However, since approach of both rewarding and aversive stimuli elicits both GABA and glutamate VP neuron activation, an additional figure (or combining information from Fig 7A and 7B) showing activation patterns of both circuits, in parallel, during reward and aversion processing could be very informative. For example: what happens in both circuits for reward vs. what happens in both circuits for aversion? This could give a broader overview of the neuronal mechanisms studied here.

We see the value in this suggestion. We have edited **Figure 7B** in response to concerns raised by all Reviewers and though we could not think of a good way to fully incorporate this idea we hope the additional discussion as described in response to Q#2 will prove satisfactory.

Minor comments

A) The authors mention that both males and females were used. In the different figures, could the authors differentiate both sexes (using different symbols when showing individual data points for example)?

Data for males and females are now disaggregated in Table S2.

B) In the Results section (last paragraph of page 9 of the pdf file), the authors wrote: “... again suggesting that VP GABA stim when evoked by the subjects’ action inhibited the resulting VTA DA neuron response”. The use of the word “inhibited” seems a little bit strong and could be replaced by a word such as “reduced” for example.

The word ‘inhibited’ has been replaced by ‘reduced’ as suggested (p.10, line 231).

C) In the Methods section, regarding “Chocolate vs. quinine-coated pellet consumption”, could the authors add a few more details on how pellets were alternatively presented? For example, did the authors presented 10 chocolate pellets and then 10 quinine-coated pellets? Or did the authors alternate between both flavors?

More details have now been added to the Methods sections regarding the Chocolate vs. quinine-coated pellet consumption p.24, lines 608–613. The Reviewer may also be interested in the added **Supplemental Figure 2** and response to Minor concern raised by Reviewer #1.

D) Throughout the manuscript the authors use “stim” to refer to optogenetic stimulation. As this does not impact the manuscript word count, could the authors use “stimulation” in the text to facilitate reading?

Agreed, we have replaced ‘stim’ with ‘stimulation’ throughout the manuscript.

Reviewer #3:

In this manuscript, Faget et al. examine the activity patterns of ventral pallidal (VP) GABA and glutamate neurons during approach and aversion, and the impact of activity in these populations on downstream activity in distinct VTA cell types, and dopamine release in subregion of the nucleus accumbens. They find that population calcium activity in both GABA and glutamate neurons in VP increases during approach to food reward, as well as a novel object (though this response decreases over time). Interestingly only GABA neurons were responsive to a cue paired with shock, whereas both cell populations responded to the shock itself. The impact of VP GABA neurons stimulation varies based on VTA cell type, as well as the animals’ control over the stimulation. The impact of VP glutamate neurons stimulation is more consistent across VTA cell types and is less sensitive to the animal’s agency. While both GABAergic and glutamatergic VP to VTA neurons can evoke increased dopamine neurons activity on a population level, glutamate neuron activation has opposing effects on dopamine release in the ventral versus dorsal subregion of the medial shell, which have been differentially implicated in reward and aversion. The data presented here are comprehensive and will be very informative to researchers trying to understand pallidal-midbrain signaling and control of motivated behavior. The statistical analysis is largely appropriate and rigorous, and the manuscript is clearly written and scholarly. I have largely minor concerns and comments, listed below.

1. In Figure 1E-F, the authors compared the magnitude of calcium activity depending on the minimal inter-lick intervals (ILIs). Larger minimal ILIs were associated with greater peak z-scored fluorescence, which they interpreted as VP GABA activity being preferentially responsive to the initial approach to an appetitive stimulus, but I'm concerned that this may be due in part to differences in the activity during the baseline period that is used for z-scoring (0-5s pre-event). It seems like licking during this baseline period when ILIs is defined as 1-2 could be artificially reducing the maximum z-score, based on a greater STD during this period. Does the pattern reported here hold true even if data is plotted as dF/F, rather than z-scored to this baseline?

We thank the Reviewer for highlighting this fair point. Please see **Figure D** for licking data expressed as dF/F (%) normalized by a 1s baseline 4 to 3s preceding the lick onset for VP GABA neurons. A similar pattern is observed with Z-score and dF/F. Please note that ILI binning strategy has been revised as detailed in response to R#1 Q#5.

2. The similar responses to both chocolate and quinine pellets in Figure 2 is somewhat surprising. Did the quinine responses change across the session as animals transitioned from consuming the pellets to only sniffing, but not consuming them?

A similar question was raised by Reviewer #1. Please see our response to Reviewer #1, first minor point, and **Supplemental Figure 2**.

3. It would be informative to know whether the cue responses in VP GABA neurons during footshock conditioning emerged following learning, or were consistent across conditioning. Was the cue response present on trial 1 (or trials 1-5)? If the cue response was present early on did it diminish at all across conditioning trials, potentially suggesting more of a novelty signal? Knowing the time course would help aid interpretation of this signal.

We thank Reviewer #3 for this comment. As requested, we extracted calcium data from trials 1, 5 and 10 of the 1st conditioning session for VP GABA neurons (see **Figure E**). VP GABA neurons presented an increase in signal during the 4s following the presentation of the 1st cue, but with a delayed time course. By the 5th and consistently until the 10th trial of the first session, this response was shifted to a rapid and transient response at cue onset.

Reviewer #4:

This work by Faget and colleagues comprises a series of comprehensive studies that advance understanding of how different populations of neurochemically-defined subtypes in the ventral pallidum orchestrate appetitive and avoidance behavior. The complementary approaches corroborate that the discrete effects of GABAergic and glutamatergic (VPGABA and VPGLu, respectively) VP neurons on appetitive behavior are driven – at least in part – by unique post-synaptic targets in the VTA, rather than the simple model previously posited that VP-GABA neurons inhibit while VP-Glu neurons excite the lateral habenula and RMTg, respectively. The dissociation of dopamine neurons

Figure D. dF/F of calcium signal at strawberry milk lick onset in VP GABA neurons (n=6).

Figure E. Z-score of calcium signal in VP GABA neurons (n=6) during 1st conditioned shock session. 1st, 5th and 10th trials are represented as average z-score \pm SEM.

projecting to the dorsal- vs. ventral-medial accumbens shell is particularly elegant in this regard. The finding that signaling in these pathways and in their ability to recruit post-synaptic targets in the VTA depends on whether actions are volitional is also novel, and is consistent with a growing body of evidence pointing to a role of the ventral pallidum in the calculation of reward prediction error.

My main suggestion for improvement refers to improving the quality of the schematic in the final figure. This serves as a summary of the many findings of the paper, and more attention to the level of detail in the circuit diagrams would help readers grasp the anatomical and functional conclusions of the manuscript.

We thank the Reviewer for their assessment of our manuscript and positive comments. We have fully revised **Figure 7** to account for this comment and those of other Reviewers.

REVIEWERS' COMMENTS

Reviewer #1 (Remarks to the Author):

The authors have addressed all my concerns. I commend their efforts to experimentally explore the relationship between reward expectancy and changes in the magnitude of VTA-DA responses to VP-GABA stimulation over time during ICSS and RTPP learning and understand the stated limitations of this approach. Nonetheless, it was great to see the difference in VTA-DA magnitude between self-stimulation and passive stimulation replicated in these preliminary data with VP-GABA to VTA terminal stimulation.

The manuscript has been improved substantially by all the additional data and analyses and it will make an excellent contribution to the field.

Reviewer #2 (Remarks to the Author):

The authors have adequately addressed all my comments.

I have only noticed a few more 'stim' in the Discussion section that could be replaced by 'stimulation'.

This is overall a very interesting study.

Reviewer #3 (Remarks to the Author):

The authors have comprehensively revised the manuscript and addressed all my prior comments.